# Uncertainty Quantification with the Empirical Neural Tangent Kernel

**Joseph Wilson**
School of Mathematics And Physics
University of Queensland
`joseph.wilson1@uqconnect.edu.au`

**Chris van der Heide**
Dept. of Electrical and Electronic Engineering
University of Melbourne
`chris.vdh@gmail.com`

**Liam Hodgkinson**
School of Mathematics and Statistics
University of Melbourne
`lhodgkinson@unimelb.edu.au`

**Fred Roosta**
CIRES and School of Mathematics And Physics
University of Queensland
`fred.roosta@uq.edu.au`

## Abstract

While neural networks have demonstrated impressive performance across various tasks, accurately quantifying uncertainty in their predictions is essential to ensure their trustworthiness and enable widespread adoption in critical systems. Several Bayesian uncertainty quantification (UQ) methods exist that are either cheap or reliable, but not both. We propose a post-hoc, sampling-based UQ method for overparameterized networks at the end of training. Our approach constructs efficient and meaningful deep ensembles by employing a (stochastic) gradient-descent sampling process on appropriately linearized networks. We demonstrate that our method effectively approximates the posterior of a Gaussian Process using the empirical Neural Tangent Kernel. Through a series of numerical experiments, we show that our method not only outperforms competing approaches in computational efficiency–often reducing costs by multiple factors–but also maintains state-of-the-art performance across a variety of UQ metrics for both regression and classification tasks.

## 1 Introduction

Neural networks (NN) achieve impressive performance on a wide array of tasks, in areas such as speech recognition (Nassif et al., 2019; Abdel-Hamid et al., 2014; Deng et al., 2013), image classification (LeCun et al., 1998; Krizhevsky et al., 2012; He et al., 2016), computer vision (Redmon et al., 2016; Redmon & Farhadi, 2017), and language processing (Vaswani et al., 2017; Ray, 2023; Devlin et al., 2019), often significantly exceeding human performance. While the promising predictive and generative performance of modern NNs is evident, accurately quantifying uncertainty in their predictions remains an important and active research frontier (Abdar et al., 2021). Models are often over-confident in predictions on out-of-distribution (OoD) inputs (Guo et al., 2017), and sensitive to distribution-shift (Ford et al., 2019). By quantifying a model's uncertainty, we can determine when it fails to provide well-calibrated predictions, indicating the need for additional training (possibly on more diverse data) or even human intervention. This is vital for deploying NNs in critical applications like diagnostic medicine and autonomous machine control (Nemani et al., 2023b).

An array of uncertainty quantification methods for NNs exist, each with benefits and drawbacks. Frequentist statistical methods, such as conformal prediction (Vovk et al., 2005; Papadopoulos et al., 2002; Lei & Wasserman, 2014), create prediction intervals through parameter estimation and probability distributions on the data. While currently considered state-of-the-art, the main drawback

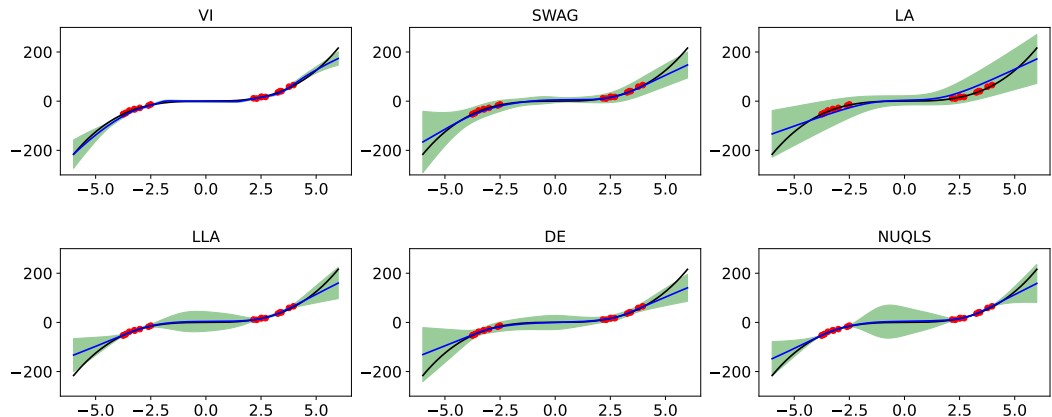

Figure 1: Comparison of various Bayesian UQ methods (see Section 2) on a 1-layer MLP, trained on the data (red) lying on $y = x^3$ (black), with Gaussian noise added. The methods' mean predictors (blue) $\pm 3\sigma$ (green) are shown, where $\sigma^2$ is the variance estimated via each method. We see that NUQLS performs well on this task.

is that conformal prediction is data-hungry, requiring a large hold-out set. Several works Altieri et al. (2024); Dadalto et al. (2023); Granese et al. (2021) employ the feature-space representations of the network to quantify the uncertainty of test predictions, through kernel densities (Kotelevskii et al., 2022), Gaussian Discriminant analysis (Mukhoti et al., 2023), non-constant mapping functions (Tagasovska & Lopez-Paz, 2019), etc. The interpretation of uncertainty in these works is related to risk of misclassification; these methods generally perform very well at detecting OoD points. While both conformal predictions and feature-space works are important, we limit our scope to Bayesian methods in this paper. Though Bayesian methods can suffer from prior misspecification (Masegosa, 2020) and computational burdens, the predictive and posterior distributions of a model are very natural frameworks for quantifying the uncertainty and spread of possible values of the model.

Unfortunately, largely due to the curse of dimensionality, existing Bayesian methods are very expensive to compute, and provide poor approximations to the predictive distribution (Folgoc et al., 2021). This results in a suite of methods that require excessive approximations to scale to large problems (Daxberger et al., 2021), often at the cost of theoretical underpinnings, or necessitating modifications to the network itself (He et al., 2020).

Gaussian Processes (GPs) (Rasmussen & Williams, 2005) are important tools in Bayesian machine learning (ML) that can directly capture the epistemic (model) uncertainty of predictions, and arise as large-width limits of NNs (Neal, 1996). However, naïve GP training scales cubically in the number of training datapoints, necessitating approximations for modern applications. Neural Tangent Kernels (NTKs) (Jacot et al., 2018) describe the functional evolution of a NN under gradient flow, and naturally arise in the analysis of model quality (Hodgkinson et al., 2023). Due to their deep connection to NNs, these covariance functions are enticing as potential tools for UQ of their corresponding NNs.

Motivated by this, we present a UQ method for a trained, over-parameterized NN model, wherein we approximate the predictive distribution through an ensemble of linearized models, trained using (stochastic) gradient-descent. For certain loss functions, this ensemble samples from the posterior of a GP with an empirical NTK.

**Contributions.** Our contributions are as follows:

1. In Section 3.1, we present a Monte-Carlo sampling based UQ method to approximate the predictive distribution of NNs, called Neural Uncertainty Quantification by Linearized Sampling (NUQLS). Our method is lightweight, post-hoc, numerically stable, and embarrassingly parallel.

2. Under certain assumptions, Section 3.2 details the convergence of NUQLS to the predictive distribution of a GP with an empirical NTK kernel, providing a novel perspective on the connection between NNs, GPs, and the NTK.

3. On various ML problems in Section 4, we show that NUQLS performs as well as or better than leading UQ methods, is less computationally expensive than deep ensemble, and scales to large image classification tasks.

**4.** In Section 4.4 we introduce a novel metric for evaluating the quality of UQ methods for classification tasks. This metric more directly measures the quality of UQ methods than existing UQ metrics.

**Remark 1.1** (Necessity of Contribution 4.). Evaluating the quality of UQ methods is non-trivial. There exists in the literature a lack of a suitable framework for evaluating the quality of UQ methods in classification settings. Common metrics such as Negative Log-Likelihood (NLL), Expected Calibration Error (ECE) (Naeini et al., 2015) and Area Under the Curve of the Receiver Operating Characteristic (AUCROC) are actually metrics for prediction quality, or are based on flawed surrogates for uncertainty. Further, it was shown in Abe et al. (2022) that the goal of Bayesian methods should not be to improve predictive ability; performance gains will be attained more economically by choosing a larger model class. Instead, Bayesian methods should seek to accurately quantify the uncertainty of a model, by computing the predictive variance. As uncertainty is not a measurement that can easily be shown to be well-calibrated, one requires a more qualitative approach to evaluating the performance of a UQ model. This is the motivation for the graphical technique we introduce in Section 4.4 for comparing the quality of UQ estimates for multi-class classification. For an in-depth discussion of these points, please see Appendix C.

**Notation.** Throughout the paper, we denote scalars, vectors, and matrices as lower-case, bold lower-case, and bold upper-case letters, e.g., $c$, $\boldsymbol{\theta}$ and $\mathbf{K}$, respectively. For two vectors $\mathbf{v} \in \mathbb{R}^p$ and $\mathbf{w} \in \mathbb{R}^p$, their Euclidean inner product is denoted as $\langle \mathbf{v}, \mathbf{w} \rangle = \mathbf{v}^\mathsf{T} \mathbf{w}$. We primarily consider supervised learning, which involves a function $\mathbf{f} : \mathbb{R}^d \times \mathbb{R}^p \to \mathbb{R}^c$, assumed sufficiently smooth in its parameters $\boldsymbol{\theta} \in \mathbb{R}^p$, a training dataset $\mathcal{D} = \{\mathcal{X}, \mathcal{Y}\} = \{\mathbf{x}_i, \mathbf{y}_i\}_{i=1}^n \subset \mathbb{R}^d \times \mathbb{R}^c$, and a loss function $\ell : \mathbb{R}^c \times \mathbb{R}^c \to [0, \infty)$.

The process of training amounts to finding a solution, $\widehat{\boldsymbol{\theta}}$, to the optimization problem $\min_{\boldsymbol{\theta}} \sum_{i=1}^n \ell(\mathbf{f}(\mathbf{x}_i, \boldsymbol{\theta}), \mathbf{y}_i) + \mathcal{R}(\boldsymbol{\theta})$, where $\mathcal{R}(\boldsymbol{\theta})$ is a regulariser. For a kernel function $\mathbf{K} : \mathbb{R}^d \times \mathbb{R}^d \to \mathbb{R}^{c \times c}$, we define $\mathbf{K}_{\mathcal{X}, \mathcal{X}} \in \mathbb{R}^{nc \times nc}$ where the $(i, j)^{\text{th}}$ (block) element is $\mathbf{K}(\mathbf{x}_i, \mathbf{x}_j) \in \mathbb{R}^{c \times c}$. Additionally, we define $\mathbf{K}_{\mathcal{X}, \mathbf{x}} \triangleq [\mathbf{K}(\mathbf{x}_1, \mathbf{x}) \quad \dots \quad \mathbf{K}(\mathbf{x}_n, \mathbf{x})]^\mathsf{T} \in \mathbb{R}^{nc \times c}$ with $\mathbf{K}_{\mathcal{X}, \mathbf{x}}^\mathsf{T} = \mathbf{K}_{\mathbf{x}, \mathcal{X}}$. For a matrix $\mathbf{J}$, its Moore-Penrose pseudo-inverse is denoted by $\mathbf{J}^\dagger$.

## 2 Background

**Bayesian Framework.** Parametric Bayesian methods admit access to a distribution over predictions $\mathbf{f}(\boldsymbol{\theta}, \mathbf{x}^\star)$ for unseen test points $\mathbf{x}^\star$, through the posterior distribution $p(\boldsymbol{\theta}|\mathcal{D}) \propto p(\boldsymbol{\theta})p(\mathcal{D}|\boldsymbol{\theta})$, and the predictive distribution $p(\mathbf{y}^\star|\mathbf{x}^\star, \mathcal{D}) = \int p(\mathbf{y}^\star|\mathbf{f}(\boldsymbol{\theta}, \mathbf{x}^\star))p(\boldsymbol{\theta}|\mathcal{D})d\boldsymbol{\theta}$, where $p(\boldsymbol{\theta})$ is the prior, and $p(\mathcal{D}|\boldsymbol{\theta})$ is the likelihood function evaluated on the training data. Both the posterior and the predictive distributions are computationally intractable in all but the simplest cases. In our setting, their calculation involves very high-dimensional integrals, which we can approximate through a Monte Carlo (MC) approximation $p(\mathbf{y}^\star|\mathbf{x}^\star, \mathcal{D}) \approx 1/S \sum_s p(\mathbf{y}^\star|\mathbf{f}(\boldsymbol{\theta}_s, \mathbf{x}^\star))$ for $\boldsymbol{\theta}_s \sim q(\boldsymbol{\theta})$, where $q(\boldsymbol{\theta})$ is an approximation to the posterior distribution. The effectiveness of classical Markov Chain Monte Carlo (MCMC) methods in posterior sampling diminishes in this setting due to the curse of dimensionality, limiting the tractable techniques available with theoretical guarantees. This limitation necessitates coarser approximations for estimating the posterior $q(\boldsymbol{\theta})$, leading to the emergence of the following Bayesian methods for posterior approximation.

Proposed in (Gal & Ghahramani, 2016), *Monte Carlo Dropout* (MC-Dropout) takes a trained NN, and uses the *dropout* regularization technique at test time to sample $S$ sub-networks, $\{\mathbf{f}(\boldsymbol{\theta}_s, \mathbf{x})\}_{s=1:S}$ as an MC approximation of the predictive distribution. MC-Dropout is an inexpensive method, yet it is unlikely to converge to the true posterior, and is erroneously multi-modal (Folgoc et al., 2021).

In *Variational Inference* (VI) (Hinton & Van Camp, 1993; Graves, 2011), a tractable family of approximating distributions for $p(\boldsymbol{\theta}|\mathcal{D})$ is chosen, denoted by $q_\psi(\boldsymbol{\theta})$, and parameterized by $\psi$. The optimal distribution in this family is obtained by finding $\psi$ that minimizes the Kullback-Leibler divergence between $q_\psi(\boldsymbol{\theta})$ and $p(\boldsymbol{\theta}|\mathcal{D})$. To be computationally viable, mean field and low-rank covariance structures are often required for $q_\psi(\boldsymbol{\theta})$.

The *Laplace Approximation* (LA) (MacKay, 1992; Ritter et al., 2018) is a tool from classical statistics which approximates the posterior distribution by a Gaussian centered at the maximum a posteriori

solution (MAP) with normalized inverse Fisher information covariance. This is justified by the Bernstein-von Mises Theorem (Van der Vaart, 2000, pp. 140–146), which guarantees that the posterior converges to this distribution in the large-data limit, for well-specified regular models. However, NNs are often over-parameterized, and the regime where $n \to \infty$ with fixed $p$ is no longer valid or a reasonable reflection of modern deep learning models (De Bortoli & Desolneux, 2022).

These limitations are acknowledged but seldom discussed by the Bayesian Deep Learning (BDL) community, which tends to view this as an additional layer of approximation rather than a modelling error, leading to the development of synonymous LA-inspired methods. However, we will show that such methods typically perform poorly compared to deep ensembles, which are often excluded from comparisons.

We can evaluate the posterior and predictive distribution in the LA using the linearization of $\mathbf{f}(\mathbf{x}, \boldsymbol{\theta})$ around the MAP solution. This approach is known as the *Linearized Laplace Approximation* (LLA) and typically delivers better performance than LA (Immer et al., 2021). LLA generally requires reduction to a subset of parameters or approximations of the covariance structure (Martens & Grosse, 2015) to scale, at the cost of performance. Recent work (Antorán et al., 2022; Ortega et al., 2023) has enabled LLA to become more scalable with better performance for larger models and datasets.

*Deep Ensembles* (DE) (Lakshminarayanan et al., 2017) are comprised of $S$ networks that are independently trained on the same training data, with different initializations, leading to a collection of parameters $\{\boldsymbol{\theta}_s; s = 1 \ldots, S\}$. At test time, our predictive distribution becomes $p(\mathbf{y}|\mathbf{x}, \mathcal{D}) \approx 1/S \sum_{s=1}^{S} p(\mathbf{y}|\mathbf{f}(\boldsymbol{\theta}_s, \mathbf{x}))$. Despite their simple construction, DEs are able to obtain samples from different modes of the posterior, and are often considered state-of-the-art for BDL (Hoffmann & Elster, 2021). However, due to the often large cost of training performant neural networks, deep ensembles of reasonable size can be undesirably expensive to obtain.

*Stochastic Weight Averaging Gaussian* (SWAG) (Maddox et al., 2019) takes a trained network and undergoes further epochs of SGD training to generate a collection of parameter samples. A Gaussian distribution with sample mean and a low-rank approximation of the sample covariance is then used to approximate a posterior mode.

**Gaussian Processes.** A GP is a stochastic process that is defined by a mean and a kernel function. A GP models the output of a random function $\mathbf{f} : \mathbb{R}^d \to \mathbb{R}^c$, at a finite collection of points $\mathbf{x}$, as being jointly Gaussian distributed. Conditioning on training data $\mathcal{D}$, it generates a posterior predictive distribution $p(\mathbf{f}(\mathbf{x}_\star)|\mathcal{D})$ at a test point $\mathbf{x}_\star$. For example, in regression settings where $c = 1$, with the mean and kernel functions $\mu : \mathbb{R}^d \to \mathbb{R}$ and $\kappa : \mathbb{R}^d \times \mathbb{R}^d \to \mathbb{R}$, as well as the observations $y \sim \mathcal{N}(f(\mathbf{x}), \sigma^2)$, there is a closed form expression for the predictive distribution, $p(f(\mathbf{x}_\star)|\mathcal{D}) \sim \mathcal{N}(\boldsymbol{\mu}(\mathbf{x}_\star), \boldsymbol{\sigma}(\mathbf{x}_\star))$, where

$$\boldsymbol{\mu}(\mathbf{x}_\star) = \mathbf{k}_{\mathbf{x}_\star, \mathcal{X}} [\mathbf{K}_{\mathcal{X}, \mathcal{X}} + \sigma^2 \mathbf{I}]^{-1} (\mathbf{y} - \boldsymbol{\mu}(\mathcal{X})) + \boldsymbol{\mu}(\mathbf{x}_\star)$$

$$\boldsymbol{\sigma}(\mathbf{x}_\star) = \kappa(\mathbf{x}_\star, \mathbf{x}_\star) - \mathbf{k}_{\mathbf{x}_\star, \mathcal{X}} [\mathbf{K}_{\mathcal{X}, \mathcal{X}} + \sigma^2 \mathbf{I}]^{-1} \mathbf{k}_{\mathcal{X}, \mathbf{x}_\star}$$

for $\mathbf{y} \triangleq [y_1, \ldots, y_n]^\mathsf{T}$ and $\boldsymbol{\mu}(\mathcal{X}) \triangleq [\mu(\mathbf{x}_1), \ldots, \mu(\mathbf{x}_n)]^\mathsf{T}$. GPs can yield impressive predictive results when a suitable kernel is chosen (Rasmussen, 1997). However, forming the kernel and solving linear systems makes GP computations intractable for large datasets. Approximations such as sparse variational inference (Titsias, 2009), Nyström methods (Martinsson & Tropp, 2020), and other subspace approximations (Gardner et al., 2018) can alleviate the computational burden; however, these approximations often result in a significant decline in predictive performance.

**Neural Tangent Kernel.** Under continuous time gradient flow, it can be shown that a NN output $\mathbf{f}(\cdot, \boldsymbol{\theta}) : \mathbb{R}^d \to \mathbb{R}^c$ undergoes kernel gradient descent, namely $\partial_t \mathbf{f}(\mathbf{x}, \boldsymbol{\theta}_t) = -\sum_{i=1}^{n} \mathbf{K}_{\boldsymbol{\theta}_t}(\mathbf{x}, \mathbf{x}_i) \nabla_{\mathbf{f}} \ell(\mathbf{f}(\mathbf{x}_i, \boldsymbol{\theta}_t), \mathbf{y}_i)$, where

$$\mathbf{K}_{\boldsymbol{\theta}}(\mathbf{x}, \mathbf{y}) \triangleq \left\langle \frac{\partial \mathbf{f}}{\partial \boldsymbol{\theta}}(\mathbf{x}, \boldsymbol{\theta}), \frac{\partial \mathbf{f}}{\partial \boldsymbol{\theta}}(\mathbf{y}, \boldsymbol{\theta}) \right\rangle \in \mathbb{R}^{c \times c}, \tag{1}$$

is the *empirical NTK* (Jacot et al., 2018). As the width of a network increases, the empirical NTK converges (in probability) to a deterministic limit, sometimes referred to as the analytic NTK, that is independent of the network's parameters. Lee et al. (2019) showed that in this limit, the network acts according to its NTK linearization during gradient descent (GD) training. This parameter

independence results in a loss of feature learning in the limiting regime (Yang & Hu, 2021). However, for finite-width NNs, Fort et al. (2020) empirically showed that the empirical NTK becomes "data-dependent" during training. Since we focus exclusively on the finite-width regime, we refer to the empirical NTK simply as the NTK.

**Related Works** Our method NUQLS shares notable similarities with, and exhibits distinct differences from, several prior works, such as Sampling-LLA, Bayesian Deep Ensembles and *local ensembles*. Due to the breadth and depth of this discussion, we relegate the related works discussion to Appendix B.

## 3 NUQLS

We now present **N**eural **U**ncertainty **Q**uantification by **L**inearized **S**ampling (NUQLS), our post-hoc sampling method for quantifying the uncertainty of a trained NN. We begin by presenting the motivation and a high-level overview of our method. Subsequently, we provide theoretical justification, demonstrating that, under specific conditions, the NUQLS samples represent draws from the approximate posterior of the neural network, which is equivalent to a GP defined by the NTK.

---

**Algorithm 1** NUQLS

**Input:** number of realizations $S$, weights $\widehat{\boldsymbol{\theta}}$.
**for** $s = 1$ **to** $S$ **do**
    $\boldsymbol{\theta}_{0,s} \leftarrow \widehat{\boldsymbol{\theta}} + \mathbf{z}_0$, where $\mathbf{z}_0 \sim \mathcal{N}(\mathbf{0}, \gamma^2 \mathbf{I})$
    $\boldsymbol{\theta}_s^{\star} \leftarrow$ Run (stochastic) GD from $\boldsymbol{\theta}_{0,s}$ to
        (approximately) solve (3) and obtain $\boldsymbol{\theta}_s^{\star}$
**end for**
**return** $\{\widetilde{\mathbf{f}}(\boldsymbol{\theta}_s^{\star}, .)\}_{s=1}^{S}$

---

### 3.1 Motivation and High-level Overview

NNs are often over-parameterized, resulting in non-uniqueness of interpolating solutions, with sub-manifolds of parameter space able to perfectly predict the training data (Hodgkinson et al., 2023). To generate a distribution over predictions, we adopt a Bayesian framework, where the uncertainty in a neural network's prediction can be interpreted as the spread of possible values the network might produce for a new test point, conditioned on the training data. To quantify this uncertainty, we can evaluate the test point on other "nearby" models with high posterior probability and analyze their range of predictions. To identify such models, we propose using the linearized approximation of the original network around its trained parameters as a simpler yet expressive surrogate. This approach can retain, to a great degree, the rich feature representation of the original network while enabling tractable exploration of the posterior distribution. In the same spirit as DE, in the overparameterized setting, we can fit this linear model to the original training data, using (stochastic) gradient descent with different initializations, resulting in an ensemble of linear predictors. Not only does this ensemble explain the training data well, but it also provides a practical way to estimate predictive uncertainty.

More precisely, let $\widehat{\boldsymbol{\theta}}$ be a set of parameters obtained after training the original NN. Linearizing $\mathbf{f}$ around $\widehat{\boldsymbol{\theta}}$ gives

$$\mathbf{f}(\boldsymbol{\theta}, \mathbf{x}) \approx \widetilde{\mathbf{f}}(\boldsymbol{\theta}, \mathbf{x}) \triangleq \mathbf{f}(\widehat{\boldsymbol{\theta}}, \mathbf{x}) + \mathbf{J}(\widehat{\boldsymbol{\theta}}, \mathbf{x})(\boldsymbol{\theta} - \widehat{\boldsymbol{\theta}}), \tag{2}$$

where $\mathbf{J}(., \mathbf{x}) = [\partial \mathbf{f}(., \mathbf{x})/\partial \boldsymbol{\theta}]^{\mathsf{T}} \in \mathbb{R}^{c \times p}$ is the Jacobian of $\mathbf{f}$. Using the linear approximation (2), we consider

$$\min_{\boldsymbol{\theta}} \sum_{i=1}^{n} \ell(\widetilde{\mathbf{f}}(\boldsymbol{\theta}, \mathbf{x}_i), \mathbf{y}_i). \tag{3}$$

In overparameterized settings, (3) may have infinitely many solutions. To identify these solutions and create our ensemble of linear predictors, we employ (stochastic) gradient descent, initialized at zero-mean isotropic Gaussian perturbations of the trained parameter, $\widehat{\boldsymbol{\theta}}$. The pseudo-code for this algorithm is provided in Algorithm 1. For a given test point $\mathbf{x}^{\star}$, the mean prediction and uncertainty can be computed using $\{\widetilde{\mathbf{f}}(\boldsymbol{\theta}_s^{\star}, \mathbf{x}^{\star})\}_{s=1}^{S}$.

Note that while the training cost for a linearised network is only slightly higher per epoch compared to standard NN training, each network in the NUQLS ensemble is initialized in a neighborhood of a local minimum of the original NN. As a result, NUQLS often requires significantly fewer epochs to converge, leading to an order-of-magnitude computational speedup relative to DE (see Tables 1, 2 and 9 for wall-clock time comparisons, and Appendix E for a more in-depth analysis of the computation costs.).

## 3.2 Theoretical Analysis

We now establish the key property of Algorithm 1: *under mild conditions, NUQLS generates samples from the approximate posterior of the neural network, which in many cases corresponds to a Gaussian process defined by the NTK*. Proofs are provided in Appendix A.

Suppose $\boldsymbol{\theta}^{\ddagger}$ is any solution to (3). Using $\boldsymbol{\theta}^{\ddagger}$, one can construct a family of solutions to (3) as

$$\boldsymbol{\theta}_{\mathbf{z}}^{\star} = \boldsymbol{\theta}^{\ddagger} + \left( \mathbf{I} - \mathbf{J}_{\mathcal{X}}^{\dagger} \mathbf{J}_{\mathcal{X}} \right) \mathbf{z}, \quad \forall \mathbf{z} \in \mathbb{R}^p, \tag{4}$$

where $\widehat{\boldsymbol{\theta}}$ is the parameters of the trained NN and $\mathbf{J}_{\mathcal{X}} = [\mathbf{J}^{\mathsf{T}}(\widehat{\boldsymbol{\theta}}, \mathbf{x}_1) \ \ldots \ \mathbf{J}^{\mathsf{T}}(\widehat{\boldsymbol{\theta}}, \mathbf{x}_n)]^{\mathsf{T}} \in \mathbb{R}^{nc \times p}$. Note that the second term in (4) consists of all vectors in the null space of $\mathbf{J}_{\mathcal{X}}$. Since any such $\boldsymbol{\theta}^{\ddagger}$ can be decomposed as the direct sum of components in the null space of $\mathbf{J}_{\mathcal{X}}$ and its orthogonal complement, the family of solutions in (4) depends on the choice of $\boldsymbol{\theta}^{\ddagger}$. However, under certain assumptions, we can ensure that the representation (4) is uniquely determined, i.e., $\boldsymbol{\theta}^{\ddagger}$ can be taken as the unique solution to (3) that is orthogonal to the null space of $\mathbf{J}_{\mathcal{X}}$. More precisely, we can show that, under these assumptions on the loss, $\boldsymbol{\theta}^{\ddagger}$ in (4) can be taken as the unique solution to

$$\min_{\boldsymbol{\theta}} \sum_{i=1}^n \ell(\widetilde{\mathbf{f}}(\boldsymbol{\theta}, \mathbf{x}_i), \mathbf{y}_i), \ \ \text{s.t.} \ \ \boldsymbol{\theta} \in \text{Range}\left( \mathbf{J}_{\mathcal{X}}^{\mathsf{T}} \right). \tag{5}$$

**Lemma 3.1.** *Suppose the loss, $\ell(\,\cdot\,, \mathbf{y})$, is either:*

- *strongly convex in its first argument, or*

- *strictly convex in its first argument, and a solution to (3) exists.*

*The problem (5) admits a unique solution.*

As it turns out, any solution of the form (4) can be efficiently obtained using (stochastic) gradient descent.

**Theorem 3.2.** *Consider the optimization problem (3) and assume $\mathbf{J}_{\mathcal{X}}$ is full row-rank.*

- *(**Gradient Descent**) Suppose $\ell(\mathbf{f}, \mathbf{y})$ is strictly convex with locally Lipschitz continuous gradient, both with respect to $\mathbf{f}$, and the problem (3) admits a solution. Gradient descent, initialized at $\mathbf{z}$ and with appropriate learning rate, converges to (4).*

- *(**Stochastic Gradient Descent**) Suppose $\ell(\mathbf{f}, \mathbf{y})$ is strongly convex with Lipschitz continuous gradient, both with respect to $\mathbf{f}$, and any solution to the problem (3) is interpolating. Stochastic gradient descent, initialized at $\mathbf{z}$ and with small enough learning rate, converges to (4) with probability one.*

We note that the local smoothness requirement in the first part of Theorem 3.2 is a relatively mild assumption; for example, it holds if we simply assume that $\ell$ is twice continuously differentiable. Also, the full-row rank assumption on $\mathbf{J}_{\mathcal{X}}$ in the second part of Theorem 3.2 is reasonable for highly over-parameterized networks; e.g., see Liu et al. (2022).

Now, suppose $\mathbf{J}_{\mathcal{X}}$ is full row-rank and the assumption of Lemma 3.1 holds, ensuring the existence of the unique solution $\boldsymbol{\theta}^{\ddagger}$ to (5). Noting $\text{Range}(\mathbf{J}_{\mathcal{X}}^{\mathsf{T}}) = \text{Range}(\mathbf{J}_{\mathcal{X}}^{\dagger})$, we can write (4) as

$$\boldsymbol{\theta}_{\mathbf{z}}^{\star} = \mathbf{J}_{\mathcal{X}}^{\mathsf{T}} \mathbf{K}_{\mathcal{X}, \mathcal{X}}^{-1} \mathbf{w} + \left( \mathbf{I} - \mathbf{J}_{\mathcal{X}}^{\mathsf{T}} \mathbf{K}_{\mathcal{X}, \mathcal{X}}^{-1} \mathbf{J}_{\mathcal{X}} \right) \mathbf{z}, \quad \forall \mathbf{z} \in \mathbb{R}^p$$

where $\mathbf{w}$ is a vector for which $\boldsymbol{\theta}^{\ddagger} = \mathbf{J}_{\mathcal{X}}^{\dagger} \mathbf{w}$, and $\mathbf{K}_{\mathcal{X}, \mathcal{X}} \triangleq \mathbf{J}_{\mathcal{X}} \mathbf{J}_{\mathcal{X}}^{\dagger} = \mathbf{K}_{\widehat{\boldsymbol{\theta}}}(\mathcal{X}, \mathcal{X}) \in \mathbb{R}^{nc \times nc}$ is the Gram matrix of the empirical NTK (1) on the training data $\mathcal{X}$. Setting $\mathbf{z} = \widehat{\boldsymbol{\theta}} - \mathbf{z}_0$ for some $\mathbf{z}_0$, we get

$$\boldsymbol{\theta}^{\star} = \mathbf{J}_{\mathcal{X}}^{\mathsf{T}} \mathbf{K}_{\mathcal{X}, \mathcal{X}}^{-1} \mathbf{w} + \left( \mathbf{I} - \mathbf{J}_{\mathcal{X}}^{\mathsf{T}} \mathbf{K}_{\mathcal{X}, \mathcal{X}}^{-1} \mathbf{J}_{\mathcal{X}} \right) (\widehat{\boldsymbol{\theta}} - \mathbf{z}_0),$$

$$\widetilde{\mathbf{f}}(\boldsymbol{\theta}^{\star}, \mathbf{x}) = \mathbf{f}(\widehat{\boldsymbol{\theta}}, \mathbf{x}) + \mathbf{K}_{\mathbf{x}, \mathcal{X}} \mathbf{K}_{\mathcal{X}, \mathcal{X}}^{-1} \left( \mathbf{w} - \mathbf{J}_{\mathcal{X}} \widehat{\boldsymbol{\theta}} \right) + \left( \mathbf{K}_{\mathbf{x}, \mathcal{X}} \mathbf{K}_{\mathcal{X}, \mathcal{X}}^{-1} \mathbf{J}_{\mathcal{X}} - \mathbf{J}(\widehat{\boldsymbol{\theta}}, \mathbf{x}) \right) \mathbf{z}_0, \tag{6}$$

where $\mathbf{K}_{\mathbf{x}, \mathcal{X}} \triangleq \mathbf{K}_{\widehat{\boldsymbol{\theta}}}(\mathbf{x}, \mathcal{X}) \in \mathbb{R}^{c \times nc}$. Taking $\mathbf{z}_0$ to be a random variable, we form an ensemble of predictors $\{\widetilde{\mathbf{f}}(\boldsymbol{\theta}^{\star}, \mathbf{x})\}_{\mathbf{z}}$, where each $\boldsymbol{\theta}^{\star}$ is formed from the projection of a random $\mathbf{z}$ onto $\text{Null}(\mathbf{J}_{\mathcal{X}})$. We require $\mathbf{z}$'s distribution to be symmetric, isotropic, and centered at $\widehat{\boldsymbol{\theta}}$, as we do not know *a priori*

which directions contain more information. We take the maximum entropy distribution for a given mean and variance, which is Gaussian[1]. Hence, we let $\mathbf{z}_0 \sim \mathcal{N}(\mathbf{0}, \gamma^2 \mathbf{I})$, for some hyper-parameter $\gamma \in \mathbb{R}$. The expectation and variance of (6), and the distribution of the predictor $\mathbf{f}(\boldsymbol{\theta}^\star, \mathbf{x})$, are then

$$\mathbb{E}\big(\widetilde{\mathbf{f}}(\boldsymbol{\theta}^\star, \mathbf{x})\big) = \boldsymbol{\mu}(\widehat{\boldsymbol{\theta}}, \mathbf{x}) = \mathbf{K}_{\mathbf{x}, \mathcal{X}}^\intercal \mathbf{K}_{\mathcal{X}, \mathcal{X}}^{-1} \mathbf{J}_{\mathcal{X}}(\boldsymbol{\theta}^\ddagger - \widehat{\boldsymbol{\theta}}) + \mathbf{f}(\widehat{\boldsymbol{\theta}}, \mathbf{x}), \tag{7}$$

$$\mathrm{Var}\big(\widetilde{\mathbf{f}}(\boldsymbol{\theta}^\star, \mathbf{x})\big) = \boldsymbol{\sigma}^2(\widehat{\boldsymbol{\theta}}, \mathbf{x}) = \big(\mathbf{K}_{\mathbf{x}, \mathbf{x}} - \mathbf{K}_{\mathbf{x}, \mathcal{X}}^\intercal \mathbf{K}_{\mathcal{X}, \mathcal{X}}^{-1} \mathbf{K}_{\mathbf{x}, \mathcal{X}}\big)\gamma^2. \tag{8}$$

$$\mathbf{f}(\boldsymbol{\theta}^\star, \mathbf{x}) \overset{\mathrm{approx}}{\sim} \mathcal{N}\big(\boldsymbol{\mu}(\widehat{\boldsymbol{\theta}}, \mathbf{x}), \boldsymbol{\sigma}^2(\widehat{\boldsymbol{\theta}}, \mathbf{x})\big). \tag{9}$$

**Remark 3.3** (Connections to GP: Regression). For scalar-valued $f : \mathbb{R}^p \times \mathbb{R}^d \to \mathbb{R}$ with quadratic loss $\ell(f(\boldsymbol{\theta}, \mathbf{x}), y) = (f(\boldsymbol{\theta}, \mathbf{x}) - y)^2$, we can explicitly write $\boldsymbol{\theta}^\ddagger = \mathbf{J}_{\mathcal{X}}^\dagger(\mathbf{y} - \mathbf{f}(\widehat{\boldsymbol{\theta}}, \mathcal{X}) + \mathbf{J}_{\mathcal{X}}\widehat{\boldsymbol{\theta}})$, where $\mathbf{f}(\boldsymbol{\theta}, \mathcal{X}) \triangleq \big[f(\boldsymbol{\theta}, \mathbf{x}_1), f(\boldsymbol{\theta}, \mathbf{x}_2), \ldots, f(\boldsymbol{\theta}, \mathbf{x}_n)\big]^\intercal$ and $\mathbf{y} \triangleq [y_1, \ldots, y_n]^\intercal$. For $\mathbf{z}_0 \sim \mathcal{N}(\mathbf{0}, \gamma^2 \mathbf{I})$, we thus have $f(\boldsymbol{\theta}^\star, \mathbf{x}) \overset{\mathrm{approx}}{\sim} \mathcal{N}\big(\mu(\widehat{\boldsymbol{\theta}}, \mathbf{x}), \sigma^2(\widehat{\boldsymbol{\theta}}, \mathbf{x})\big)$ with

$$\mu(\widehat{\boldsymbol{\theta}}, \mathbf{x}) = \mathbf{k}_{\mathbf{x}, \mathcal{X}} \mathbf{K}_{\mathcal{X}, \mathcal{X}}^{-1}(\mathbf{y} - \mathbf{f}(\widehat{\boldsymbol{\theta}}, \mathcal{X})) + f(\widehat{\boldsymbol{\theta}}, \mathbf{x}),$$

$$\sigma^2(\widehat{\boldsymbol{\theta}}, \mathbf{x}) = \big(\kappa(\mathbf{x}, \mathbf{x}) - \mathbf{k}_{\mathbf{x}, \mathcal{X}} \mathbf{K}_{\mathcal{X}, \mathcal{X}}^{-1} \mathbf{k}_{\mathbf{x}, \mathcal{X}}\big)\gamma^2.$$

By the full-rank assumption on the Jacobian, this amounts to the conditional distribution of the following normal distribution, conditioned on interpolation $\mathbf{f}(\boldsymbol{\theta}^\star, \mathcal{X}) = \mathbf{y}$,

$$\begin{bmatrix} \mathbf{f}(\boldsymbol{\theta}^\star, \mathcal{X}) \\ f(\boldsymbol{\theta}^\star, \mathbf{x}) \end{bmatrix} \sim \mathcal{N}\left( \begin{bmatrix} \mathbf{f}(\widehat{\boldsymbol{\theta}}, \mathcal{X}) \\ f(\widehat{\boldsymbol{\theta}}, \mathbf{x}) \end{bmatrix}, \gamma^2 \begin{bmatrix} \mathbf{K}_{\mathcal{X}, \mathcal{X}} & \mathbf{k}_{\mathbf{x}, \mathcal{X}} \\ \mathbf{k}_{\mathbf{x}, \mathcal{X}}^\intercal & \kappa_{\mathbf{x}, \mathbf{x}} \end{bmatrix} \right). \tag{10}$$

Therefore, $f(\boldsymbol{\theta}^\star, \mathbf{x})$ follows a GP with an NTK kernel. Conditioning on $\mathbf{f}(\boldsymbol{\theta}^\star, \mathcal{X}) = \mathbf{y}$ is reasonable, since by construction $\mathbf{f}(\boldsymbol{\theta}^\star, \mathcal{X}) \approx \widetilde{\mathbf{f}}(\boldsymbol{\theta}^\star, \mathcal{X})$, and under the full-rank assumption of $\mathbf{J}_{\mathcal{X}}$, we have $\widetilde{\mathbf{f}}(\boldsymbol{\theta}^\star, \mathcal{X}) = \mathbf{y}$.

**Remark 3.4** (Connections to GP: General Loss). Beyond quadratic loss, for a general loss function satisfying the assumptions of Lemma 3.1, a clear GP posterior interpretation like that in (10) may not exist. Nevertheless, we can still derive related insights. If $\widehat{\boldsymbol{\theta}}$ is an interpolating solution from initial training, which is common for modern NNs, then as long as $\mathbf{J}_{\mathcal{X}}$ is full row-rank, solving (3) is equivalent to finding $\boldsymbol{\theta} \in \mathbb{R}^p$ such that $\widetilde{\mathbf{f}}(\boldsymbol{\theta}, \mathcal{X}) = \mathbf{f}(\widehat{\boldsymbol{\theta}}, \mathcal{X})$, i.e., find $\boldsymbol{\theta} \in \mathbb{R}^p$ such that $\mathbf{J}_{\mathcal{X}}(\boldsymbol{\theta} - \widehat{\boldsymbol{\theta}}) = \mathbf{0}$. So we obtain (9) with $\boldsymbol{\mu}(\widehat{\boldsymbol{\theta}}, \mathbf{x}) = \mathbf{f}(\widehat{\boldsymbol{\theta}}, \mathbf{x})$ and $\boldsymbol{\sigma}^2(\widehat{\boldsymbol{\theta}}, \mathbf{x}) = \big(\mathbf{K}_{\mathbf{x}, \mathbf{x}} - \mathbf{K}_{\mathbf{x}, \mathcal{X}}^\intercal \mathbf{K}_{\mathcal{X}, \mathcal{X}}^{-1} \mathbf{K}_{\mathbf{x}, \mathcal{X}}\big)\gamma^2$ as the conditional distribution of (10), conditioned on the event $\mathbf{f}(\boldsymbol{\theta}^\star, \mathcal{X}) = \mathbf{f}(\widehat{\boldsymbol{\theta}}, \mathcal{X})$, i.e., interpolation with $\ell(\mathbf{f}(\boldsymbol{\theta}^\star, \mathbf{x}_i), \mathbf{y}_i) = 0$ for $i = 1, \ldots, n$.

**The Punchline.** Drawing samples from the posterior (9) by explicitly calculating (7) and (8) can be intractable in large-scale settings. Moreover, it can be numerically unstable due to the highly ill-conditioned nature of the NTK matrix[2]. However, by combining the above derivations with Theorem 3.2, we arrive at the key property of Algorithm 1: it enables efficient sampling from the posterior (9).

**Corollary 3.5** (Key Property of NUQLS). *With the assumptions of Theorem 3.2, the samples generated by Algorithm 1 represent draws from the predictive distribution in* (9).

Hence, we approximate (7) and (8) by computing the sample mean and covariance of $\{\widetilde{\mathbf{f}}(\boldsymbol{\theta}_s^\star, \mathbf{x})\}_{s=1}^S$ obtained from Algorithm 1. By the law of large numbers, the quality of these approximations improves as $S \to \infty$.

**Remark 3.6.** Loss functions that do not satisfy the assumption of Lemma 3.1, such as the cross-entropy loss, may fail to yield a unique representation of (4), so the above posterior analysis does not apply. However, our experiments demonstrate that Algorithm 1 can still generate samples that effectively capture the posterior variance (see Section 4.4) and posterior mean (see Appendix G.1). In cases where (3) lacks a solution, Algorithm 1 can still be executed by terminating the iterations of (stochastic) GD early. Investigating the distribution of the resulting ensemble and its connection to an explicit posterior remains a potential direction for future research.

---

[1]Heavier-tailed distributions matching the above criteria, e.g. logistic distributions, *may* improve results.

[2]Recall that the condition number of $\mathbf{K}_{\mathcal{X}, \mathcal{X}}$ is the square of that of $\mathbf{J}_{\mathcal{X}}$.

# 4 Experiments

We now empirically demonstrate the result of Theorem 3.2, as well as compare the performance of our method with alternatives on various regression and classification tasks. Implementation details are given in Appendix I. The PyTorch implementation of our experiments is available here. We have also released our method as a package. For additional experimental results, please see Appendix H.

## 4.1 Empirical Convergence

We demonstrate empirically the convergence of the predictive distribution of NUQLS to that of an NTK-GP. We take a series of MLPs with NTK scaling (Jacot et al., 2018), and train these NNs on normalized Gaussian data (a regression task). Under this setting, the condition number of the resultant NTK is reasonable ($\approx 10^3 - 10^8$, compared to $\approx 10^{17}$ on some UCI regression datasets). This allows us to explicitly compute the predictive variance for the NTK-GP for some normalized Gaussian test set. We then also compute the predictive variance using NUQLS, and take the $l_2$ norm of the difference between the two variance sets, which we term Squared Error of the Variance (SEV). We plot this against number of epochs of training for NUQLS, and the number of NUQLS realizations $s$. We also plot the average training loss of the linearised networks in the NUQLS ensemble. The values are averaged over $10$ random realizations. The results are displayed in Figure 2. Note that the SEV cannot be zero, due to the condition number of the NTK; this affects the solution of the linear system $K_{XX}\mathbf{y} = K_{Xx}$ in the computation of the

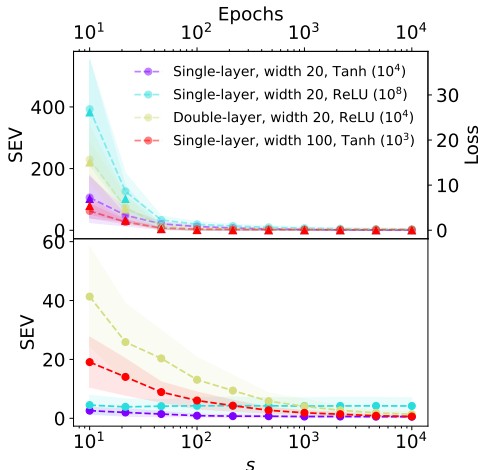

Figure 2: Plot of SEV (●) and NUQLS train loss (▲) against (top) number of epochs of training for NUQLS and (bottom) number of NUQLS realizations. Bracketed number is condition number of NTK Gram matrix. Mean and $95\%$ confidence intervals are shown from $10$ random realizations.

variance for the NTK-GP. However, we see clear convergence of our method to the distribution of an NTK-GP as the ensemble members of NUQLS approach their minima, and as the number of ensemble members increases.

## 4.2 Toy Regression

We compare the performance of our method on a toy regression problem, taken from (Hernández-Lobato & Adams, 2015) and extended in (Park & Blei, 2024). In Figure 1, we take 20 uniformly sampled points in the domain $x \in [-4, -2] \cup [2, 4]$, and let $y = x^3 + \epsilon$, $\epsilon \sim \mathcal{N}(0, 3^2)$. A small MLP was trained on these data and used for prediction. We apply VI, SWAG, LA, LLA, DE and NUQLS to the network to find a predictive mean and uncertainty. Close to the training data, i.e. in $[-4, -2] \cup [2, 4]$ we expect low uncertainty; outside of this region, the uncertainty should grow with distance from the training points. VI underestimates, while SWAG and LA overestimate the uncertainty. DE grows more uncertain with distance from the training points, however both NUQLS and the LLA contain the underlying target curve within their confidence intervals. Note that deep ensembles output a heteroskedastic variance term, and were trained on a Gaussian likelihood; in comparison, the variances for LLA and NUQLS were computed post-hoc.

## 4.3 UCI Regression

In Tables 1 and 9, we compare NUQLS with DE, LLA and SWAG on a series of UCI regression problems. Mean squared error (MSE) and expected calibration error (ECE) respectively evaluate the predictive and UQ performance, with Gaussian negative log likelihood (NLL) evaluating both. See (Nemani et al., 2023a, §4.11) for an explanation of ECE. We see that NUQLS consistently has the (equal) best ECE (except for the Song dataset, where it falls short of the LLA ECE by $0.1\%$). It has comparable or better NLL than other methods on all datasets, and often gives an improvement on RMSE. Finally, it is the quickest method, often by a very significant margin, and it does not *fail*

Table 1: Comparing performance of NUQLS, DE, LLA and SWAG on UCI regression tasks. NUQLS performs as well as or better than all other methods, while showing a speed up over other methods; this speed up increases with the size of the datasets. LLA-K denotes LLA with a KFAC covariance structure. Reported time for NUQLS, LLA and SWAG includes the training time for the original NN, with the run-time of the post-hoc method given in brackets.

| Dataset | Method | RMSE ↓ | NLL ↓ | ECE ↓ | Time(s) |
|---|---|---|---|---|---|
| **Energy** | NUQLS | $\mathbf{0.047}_{\pm 0.006}$ | $\mathbf{-2.400}_{\pm 0.209}$ | $\mathbf{0.002}_{\pm 0.002}$ | $\mathbf{8.374\ (0.151)}$ |
| | DE | $0.218_{\pm 0.032}$ | $\mathbf{-1.651}_{\pm 0.783}$ | $0.004_{\pm 0.002}$ | $102.244$ |
| | LLA | $\mathbf{0.048}_{\pm 0.006}$ | $\mathbf{-2.475}_{\pm 0.128}$ | $0.004_{\pm 0.004}$ | $8.491\ (0.269)$ |
| | SWAG | $0.058_{\pm 0.015}$ | $-1.950_{\pm 0.158}$ | $0.080_{\pm 0.011}$ | $45.306\ (37.084)$ |
| **Kin8nm** | NUQLS | $\mathbf{0.252}_{\pm 0.005}$ | $-0.796_{\pm 0.025}$ | $\mathbf{0.000}_{\pm 0.000}$ | $\mathbf{26.570\ (0.264)}$ |
| | DE | $\mathbf{0.252}_{\pm 0.006}$ | $\mathbf{-0.914}_{\pm 0.028}$ | $0.002_{\pm 0.001}$ | $73.967$ |
| | LLA | $0.260_{\pm 0.010}$ | $-0.783_{\pm 0.054}$ | $0.001_{\pm 0.001}$ | $38.272\ (11.966)$ |
| | SWAG | $0.457_{\pm 0.149}$ | $-0.006_{\pm 0.295}$ | $0.054_{\pm 0.012}$ | $176.569\ (150.263)$ |
| **Protein** | NUQLS | $\mathbf{0.623}_{\pm 0.005}$ | $0.209_{\pm 0.047}$ | $\mathbf{0.002}_{\pm 0.000}$ | $\mathbf{81.264\ (1.356)}$ |
| | DE | $0.741_{\pm 0.052}$ | $\mathbf{0.203}_{\pm 0.203}$ | $0.011_{\pm 0.020}$ | $1014.827$ |
| | LLA-K | $0.640_{\pm 0.007}$ | $0.458_{\pm 0.071}$ | $\mathbf{0.002}_{\pm 0.000}$ | $89.414\ (9.506)$ |
| | SWAG | $0.730_{\pm 0.044}$ | $\mathbf{0.187}_{\pm 0.080}$ | $\mathbf{0.002}_{\pm 0.002}$ | $548.88\ (468.972)$ |

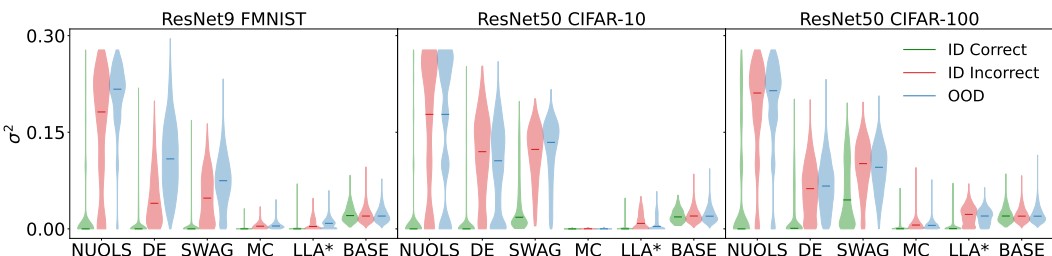

Figure 3: Violin plot of VMSP, for correctly predicted ID test points, incorrectly predicted ID test points, and OoD test points. Median is shown, with violin width depicting density. Low variance is expected for ID correct points, and large variance for ID incorrect and OoD points.

on any datasets, like the other methods do. Note that for the two largest datasets, Protein and Song, we required approximations on the covariance structure of LLA (see (Daxberger et al., 2021)). A detailed explanation of the hyper-parameter tuning method for NUQLS is given in Appendix F.1.

## 4.4 Image Classification - Uncertainty

We now compare the UQ performance of NUQLS, DE, SWAG, LLA* (LLA with a last-layer and KFAC approximation), and MC-Dropout (MC), on larger image classification tasks. We take variance of the maximum predicted softmax probability (VMSP), for a given test point, as the correct quantifier of uncertainty in this setting (see Appendix C for justification).

Figure 3 presents a violin plot of the VMSP for three test-groups: correctly predicted in-distribution (FashionMNIST, CIFAR-10) test points, incorrectly predicted in-distribution test points, and out-of-distribution (MNIST, CIFAR-100) test points. We would expect that there should be, on average, much smaller uncertainty for ID test points that have been correctly predicted, and larger uncertainty for incorrectly predicted ID and OoD test points. We compare against a completely randomized baseline method (BASE), where we sample 10 standard normal realizations of logits, passed through a softmax. In Table 10 in Appendix H.8, we display the corresponding median and sample skew values for each method in each test group, to quantify the distribution of VMSP for each test set. Ideally, a method should far outperform the baseline, so we can use the median and sample skew difference between a method and the baseline as a way to compare different methods. We see that NUQLS outperforms all other methods, including the SOTA method DE. In Appendix H.7 we provide additional experimental evaluation of NUQLS. In Figure 6 we evaluate NUQLS on a ResNet50 trained on both SVHN and ImageNet, displaying the scalability of our method, as well as providing comparison with other competing methods, including Bayesian Deep Ensembles (BDE), Spectral-Normalized Neural Gaussian Process (SNGP), BatchEnsemble (BE), and Stochastic

Gradient Langevin Dynamics (SGLD). Against these competing methods, NUQLS performs the strongest.

## 5 Conclusion

We have presented NUQLS, a Bayesian, post-hoc UQ method that approximates the predictive distribution of an over-parameterized NN through GD/SGD training of linear networks, allowing scalability without sacrificing performance. Under assumptions on the loss function, this predictive distribution reduces to a GP using the NTK. We find that our method is competitive with, and often far outperforms, existing UQ methods on regression and classification tasks, whilst providing a novel connection between NNs, GPs and the NTK.

**Limitation.** A theoretical limitation of this work is that its connection to the NTK-GP does not extend to loss functions that violate the assumptions of Theorem 3.2, such as the cross-entropy loss. The strong empirical performance of NUQLS on classification tasks motivates future research to extend Theorem 3.2 to broader classes of loss functions and alternative optimization algorithms. A further limitation of the method is the dependence of NUQLS on the linearization approximation. As can be seen in Appendix H.1, when the target neural network is poorly trained, and hence the loss-landscape is far from flat around the 'trained' parameters, the performance of NUQLS suffers. However, when the network is well-trained, as is common in practical settings, the linear approximation holds and hence NUQLS performs well.

## Acknowledgments and Disclosure of Funding

Liam Hodgkinson is supported by the Australian Research Council through a Discovery Early Career Researcher Award (DE240100144). Fred Roosta was partially supported by the Australian Research Council through an Industrial Transformation Training Centre for Information Resilience (IC200100022).

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

# A  Proofs

*Proof of Lemma 3.1.* First, we note that by the assumption on $\ell$, a solution to (5) always exists. Suppose to the contrary that (5) has two distinct solutions $\tilde{\boldsymbol{\theta}}$ and $\widehat{\boldsymbol{\theta}}$ such that $\tilde{\boldsymbol{\theta}} \neq \widehat{\boldsymbol{\theta}}$. Since $\tilde{\boldsymbol{\theta}} \in$ Range$([\mathbf{J}(\widehat{\boldsymbol{\theta}}, \mathcal{X})]^{\mathsf{T}})$ and $\widehat{\boldsymbol{\theta}} \in$ Range$([\mathbf{J}(\widehat{\boldsymbol{\theta}}, \mathcal{X})]^{\mathsf{T}})$, i.e., $(\tilde{\boldsymbol{\theta}} - \widehat{\boldsymbol{\theta}}) \perp$ Null$([\mathbf{J}(\widehat{\boldsymbol{\theta}}, \mathcal{X})])$, it follows that $\mathbf{J}(\widehat{\boldsymbol{\theta}}, \mathcal{X})\tilde{\boldsymbol{\theta}} \neq \mathbf{J}(\widehat{\boldsymbol{\theta}}, \mathcal{X})\widehat{\boldsymbol{\theta}}$, which in particular implies $\langle \nabla \mathbf{f}(\widehat{\boldsymbol{\theta}}, \mathbf{x}_i), \tilde{\boldsymbol{\theta}} \rangle \neq \langle \nabla \mathbf{f}(\widehat{\boldsymbol{\theta}}, \mathbf{x}_i), \widehat{\boldsymbol{\theta}} \rangle$ for all $i = 1, \ldots, n$. Consider $\bar{\boldsymbol{\theta}} = (\tilde{\boldsymbol{\theta}} + \widehat{\boldsymbol{\theta}})/2$. By strict convexity on $\ell$, we have

$$
\begin{aligned}
\sum_{i=1}^{n} \ell(\widetilde{\mathbf{f}}(\bar{\boldsymbol{\theta}}, \mathbf{x}_i), y_i) &= \sum_{i=1}^{n} \ell\left( \mathbf{f}(\widehat{\boldsymbol{\theta}}, \mathbf{x}_i) + \left\langle \nabla \mathbf{f}(\widehat{\boldsymbol{\theta}}, \mathbf{x}_i), \frac{\tilde{\boldsymbol{\theta}} + \widehat{\boldsymbol{\theta}}}{2} - \widehat{\boldsymbol{\theta}} \right\rangle, y_i \right) \\
&= \sum_{i=1}^{n} \ell\left( \frac{\mathbf{f}(\widehat{\boldsymbol{\theta}}, \mathbf{x}_i) + \left\langle \nabla \mathbf{f}(\widehat{\boldsymbol{\theta}}, \mathbf{x}_i), \tilde{\boldsymbol{\theta}} - \widehat{\boldsymbol{\theta}} \right\rangle}{2} + \frac{\mathbf{f}(\widehat{\boldsymbol{\theta}}, \mathbf{x}_i) + \left\langle \nabla \mathbf{f}(\widehat{\boldsymbol{\theta}}, \mathbf{x}_i), \widehat{\boldsymbol{\theta}} - \widehat{\boldsymbol{\theta}} \right\rangle}{2}, y_i \right) \\
&< \frac{1}{2} \sum_{i=1}^{n} \ell\left( \widetilde{\mathbf{f}}(\tilde{\boldsymbol{\theta}}, \mathbf{x}_i), y_i \right) + \frac{1}{2} \sum_{i=1}^{n} \ell\left( \widetilde{\mathbf{f}}(\widehat{\boldsymbol{\theta}}, \mathbf{x}_i), y_i \right) \\
&= \sum_{i=1}^{n} \ell\left( \widetilde{\mathbf{f}}(\tilde{\boldsymbol{\theta}}, \mathbf{x}_i), y_i \right),
\end{aligned}
$$

which is a contradiction. $\qquad\square$

*Proof of Theorem 3.2.*

- (Gradient Descent) Denoting

$$
\mathbf{J}(\widehat{\boldsymbol{\theta}}, \mathcal{X}) \triangleq \left[ \mathbf{J}^{\mathsf{T}}(\widehat{\boldsymbol{\theta}}, \mathbf{x}_1) \ \ldots \ \mathbf{J}^{\mathsf{T}}(\widehat{\boldsymbol{\theta}}, \mathbf{x}_n) \right]^{\mathsf{T}} \in \mathbb{R}^{nc \times p},
$$

we can write

$$
\mathbf{z} = \mathbf{J}^{\dagger}(\widehat{\boldsymbol{\theta}}, \mathcal{X})\mathbf{J}(\widehat{\boldsymbol{\theta}}, \mathcal{X})\mathbf{z} + \left( \mathbf{I} - \mathbf{J}^{\dagger}(\widehat{\boldsymbol{\theta}}, \mathcal{X})\mathbf{J}(\widehat{\boldsymbol{\theta}}, \mathcal{X}) \right) \mathbf{z},
$$

where $\widehat{\boldsymbol{\theta}}$ represents the parameters around which the linear model $\widetilde{\mathbf{f}}$ is defined in (2). The first iteration of gradient descent, initialized at $\boldsymbol{\theta}^{(0)} = \mathbf{z}$, is given by

$$
\begin{aligned}
\boldsymbol{\theta}^{(1)} &= \mathbf{z} - \alpha \sum_{i=1}^{n} \frac{\partial \widetilde{\mathbf{f}}}{\partial \boldsymbol{\theta}}(\mathbf{z}, \mathbf{x}_i) \nabla \ell(\widetilde{\mathbf{f}}(\mathbf{z}, \mathbf{x}_i), \mathbf{y}_i) \\
&= \left( \mathbf{I} - \mathbf{J}^{\dagger}(\widehat{\boldsymbol{\theta}}, \mathcal{X})\mathbf{J}(\widehat{\boldsymbol{\theta}}, \mathcal{X}) \right) \mathbf{z} + \mathbf{J}^{\dagger}(\widehat{\boldsymbol{\theta}}, \mathcal{X})\mathbf{J}(\widehat{\boldsymbol{\theta}}, \mathcal{X})\mathbf{z} - \alpha \sum_{i=1}^{n} \frac{\partial \widetilde{\mathbf{f}}}{\partial \boldsymbol{\theta}}(\mathbf{z}, \mathbf{x}_i) \nabla \ell(\widetilde{\mathbf{f}}(\mathbf{z}, \mathbf{x}_i), \mathbf{y}_i) \\
&= \left( \mathbf{I} - \mathbf{J}^{\dagger}(\widehat{\boldsymbol{\theta}}, \mathcal{X})\mathbf{J}(\widehat{\boldsymbol{\theta}}, \mathcal{X}) \right) \mathbf{z} + \mathbf{J}^{\dagger}(\widehat{\boldsymbol{\theta}}, \mathcal{X})\mathbf{J}(\widehat{\boldsymbol{\theta}}, \mathcal{X})\mathbf{z} - \alpha \sum_{i=1}^{n} \left[ \mathbf{J}(\widehat{\boldsymbol{\theta}}, \mathbf{x}_i) \right]^{\mathsf{T}} \nabla \ell(\widetilde{\mathbf{f}}(\mathbf{z}, \mathbf{x}_i), \mathbf{y}_i) \\
&= \left( \mathbf{I} - \mathbf{J}^{\dagger}(\widehat{\boldsymbol{\theta}}, \mathcal{X})\mathbf{J}(\widehat{\boldsymbol{\theta}}, \mathcal{X}) \right) \mathbf{z} + \mathbf{v}^{(1)},
\end{aligned}
$$

where $\mathbf{v}^{(1)} \in$ Range$\left( \left[ \mathbf{J}(\widehat{\boldsymbol{\theta}}, \mathcal{X}) \right]^{\mathsf{T}} \right)$. The next iteration is similarly given by

$$
\begin{aligned}
\boldsymbol{\theta}^{(2)} &= \boldsymbol{\theta}^{(1)} - \alpha \sum_{i=1}^{n} \frac{\partial \widetilde{\mathbf{f}}}{\partial \boldsymbol{\theta}}(\boldsymbol{\theta}^{(1)}, \mathbf{x}_i) \nabla \ell(\widetilde{\mathbf{f}}(\boldsymbol{\theta}^{(1)}, \mathbf{x}_i), \mathbf{y}_i) \\
&= \left( \mathbf{I} - \mathbf{J}^{\dagger}(\widehat{\boldsymbol{\theta}}, \mathcal{X})\mathbf{J}(\widehat{\boldsymbol{\theta}}, \mathcal{X}) \right) \mathbf{z} + \mathbf{v}^{(1)} - \alpha \sum_{i=1}^{n} \left[ \mathbf{J}(\widehat{\boldsymbol{\theta}}, \mathbf{x}_i) \right]^{\mathsf{T}} \nabla \ell(\widetilde{\mathbf{f}}(\boldsymbol{\theta}^{(1)}, \mathbf{x}_i), \mathbf{y}_i) \\
&= \left( \mathbf{I} - \mathbf{J}^{\dagger}(\widehat{\boldsymbol{\theta}}, \mathcal{X})\mathbf{J}(\widehat{\boldsymbol{\theta}}, \mathcal{X}) \right) \mathbf{z} + \mathbf{v}^{(1)} + \mathbf{v}^{(2)},
\end{aligned}
$$

where again $\mathbf{v}^{(2)} \in \text{Range}\left(\left[\mathbf{J}(\widehat{\boldsymbol{\theta}}, \mathcal{X})\right]^{\intercal}\right)$. Generalizing to the $n$th iteration,

$$\boldsymbol{\theta}^{(n)} = \left(\mathbf{I} - \mathbf{J}^\dagger(\widehat{\boldsymbol{\theta}}, \mathcal{X})\mathbf{J}(\widehat{\boldsymbol{\theta}}, \mathcal{X})\right)\mathbf{z} + \sum_{i=1}^{n}\mathbf{v}^{(i)}.$$

Hence, by the assumption on $\ell$, as long as an adaptive learning rate is chosen appropriately according to Malitsky & Mishchenko (2020), GD must converge to a solution of the form

$$\boldsymbol{\theta}^\star = \left(\mathbf{I} - \mathbf{J}^\dagger(\widehat{\boldsymbol{\theta}}, \mathcal{X})\mathbf{J}(\widehat{\boldsymbol{\theta}}, \mathcal{X})\right)\mathbf{z} + \sum_{i=1}^{\infty}\mathbf{v}^{(i)}$$

$$= \left(\mathbf{I} - \mathbf{J}^\dagger(\widehat{\boldsymbol{\theta}}, \mathcal{X})\mathbf{J}(\widehat{\boldsymbol{\theta}}, \mathcal{X})\right)\mathbf{z} + \mathbf{v},$$

where $\mathbf{v} \in \text{Range}\left(\left[\mathbf{J}(\widehat{\boldsymbol{\theta}}, \mathcal{X})\right]^{\intercal}\right)$. In particular, for $\mathbf{z} = \mathbf{0}$, by Lemma 3.1, we must have that $\mathbf{v} = \boldsymbol{\theta}^\ddagger$, where $\boldsymbol{\theta}^\ddagger$ is the solution to (5). Therefore,

$$\boldsymbol{\theta}^\star = \left(\mathbf{I} - \mathbf{J}^\dagger(\widehat{\boldsymbol{\theta}}, \mathcal{X})\mathbf{J}(\widehat{\boldsymbol{\theta}}, \mathcal{X})\right)\mathbf{z} + \boldsymbol{\theta}^\ddagger.$$

- (Stochastic Gradient Descent) Using a similar argument as above, it is easy to show that each iteration of the mini-batch SGD is of the form

$$\boldsymbol{\theta}^{(n)} \in \left(\mathbf{I} - \mathbf{J}^\dagger(\widehat{\boldsymbol{\theta}}, \mathcal{X})\mathbf{J}(\widehat{\boldsymbol{\theta}}, \mathcal{X})\right)\mathbf{z} + \text{Range}\left(\left[\mathbf{J}(\widehat{\boldsymbol{\theta}}, \mathcal{X})\right]^{\intercal}\right).$$

Hence, it suffices to show that SGD converges almost surely. Defining

$$\mathcal{L}(\boldsymbol{\theta}) \triangleq \sum_{i=1}^{n}\ell(\widetilde{\mathbf{f}}(\boldsymbol{\theta}, \mathbf{x}_i), \mathbf{y}_i), \quad \text{and} \quad \mathbf{g}(\boldsymbol{\theta}, \mathcal{X}) \triangleq \begin{bmatrix}\nabla\ell(\widetilde{\mathbf{f}}(\boldsymbol{\theta}, \mathbf{x}_1), \mathbf{y}_1) \\ \vdots \\ \nabla\ell(\widetilde{\mathbf{f}}(\boldsymbol{\theta}, \mathbf{x}_n), \mathbf{y}_n)\end{bmatrix} \in \mathbb{R}^{nc},$$

we write

$$\nabla\mathcal{L}(\boldsymbol{\theta}) = \sum_{i=1}^{n}\left[\mathbf{J}(\widehat{\boldsymbol{\theta}}, \mathbf{x}_1)\right]^{\intercal}\nabla\ell(\widetilde{\mathbf{f}}(\boldsymbol{\theta}, \mathbf{x}_i), \mathbf{y}_i) = \left[\mathbf{J}(\widehat{\boldsymbol{\theta}}, \mathcal{X})\right]^{\intercal}\mathbf{g}(\boldsymbol{\theta}, \mathcal{X}).$$

Let $\boldsymbol{\theta}^\star$ be any solution to (3). The full row-rank assumption on $\mathbf{J}(\widehat{\boldsymbol{\theta}}, \mathcal{X})$ implies that we must have $\mathbf{g}(\boldsymbol{\theta}^\star, \mathcal{X}) = \mathbf{0}$, i.e., $\nabla\ell(\widetilde{\mathbf{f}}(\boldsymbol{\theta}^\star, \mathbf{x}_i), \mathbf{y}_i) = \mathbf{0}$, $i = 1, \ldots, n$. By the $\mu$-strong convexity assumption on $\ell(., \mathbf{y})$ with respect to its first argument, it is easy to see that $\mathcal{L}(\boldsymbol{\theta})$ satisfies the Polyak-Łojasiewicz inequality (Karimi et al., 2016) with constant $2\mu\lambda$ where

$$\lambda \triangleq \min_{i=1,\ldots,n}\sigma_{\min}^2(\mathbf{J}(\widehat{\boldsymbol{\theta}}, \mathbf{x}_i)),$$

and $\sigma_{\min}(\mathbf{J}(\widehat{\boldsymbol{\theta}}, \mathbf{x}_i))$ is the smallest non-zero singular value of $\mathbf{J}(\widehat{\boldsymbol{\theta}}, \mathbf{x}_i)$. Indeed, let $\boldsymbol{\theta}^\star$ be any solution to (3). From $\mu$-strong convexity of $\ell(., \mathbf{y})$ with respect to its first argument, for any $\boldsymbol{\theta}$, we have

$$\ell(\widetilde{\mathbf{f}}(\boldsymbol{\theta}, \mathbf{x}_i), \mathbf{y}_i) - \ell(\widetilde{\mathbf{f}}(\boldsymbol{\theta}^\star, \mathbf{x}_i), \mathbf{y}_i) \leq \frac{1}{2\mu}\left\|\nabla\ell(\widetilde{\mathbf{f}}(\boldsymbol{\theta}, \mathbf{x}_i), \mathbf{y}_i)\right\|^2$$

$$\leq \frac{1}{2\mu\sigma_{\min}(\mathbf{J}(\widehat{\boldsymbol{\theta}}, \mathbf{x}_i))}\left\|\left[\mathbf{J}(\widehat{\boldsymbol{\theta}}, \mathbf{x}_i)\right]^{\intercal}\nabla\ell(\widetilde{\mathbf{f}}(\boldsymbol{\theta}, \mathbf{x}_i), \mathbf{y}_i)\right\|^2,$$

which implies

$$\mathcal{L}(\boldsymbol{\theta}) - \mathcal{L}(\boldsymbol{\theta}^\star) \leq \frac{1}{2\mu\lambda}\left\|\left[\mathbf{J}(\widehat{\boldsymbol{\theta}}, \mathcal{X})\right]^{\intercal}\mathbf{g}(\boldsymbol{\theta}, \mathcal{X})\right\|^2 = \frac{1}{2\mu\lambda}\left\|\nabla\mathcal{L}(\boldsymbol{\theta})\right\|^2.$$

By the smoothness assumption on each $\ell(\widetilde{\mathbf{f}}(., \mathbf{x}_i), \mathbf{y}_i)$ as well as the interpolating property of $\boldsymbol{\theta}^{\star}$, Bassily et al. (2018, Theorem 1) implies that the mini-batch SGD with small enough step size $\eta$ has an exponential convergence rate as

$$\mathbb{E}\mathcal{L}(\boldsymbol{\theta}^{(k)}) \leq (1-\rho)^k \mathcal{L}(\boldsymbol{\theta}^{(0)}),$$

for some contact $0 < \rho < 1$. This in particular implies that for any $\epsilon > 0$,

$$\sum_{k=1}^{\infty} \mathbb{P}(\mathcal{L}(\boldsymbol{\theta}^{(k)}) > \epsilon) \leq \sum_{k=1}^{\infty} \frac{\mathbb{E}\mathcal{L}(\boldsymbol{\theta}^{(k)})}{\epsilon} \leq \frac{\mathcal{L}(\boldsymbol{\theta}^{(0)})}{\epsilon} \sum_{k=1}^{\infty} (1-\rho)^k = \frac{\mathcal{L}(\boldsymbol{\theta}^{(0)})}{\epsilon\rho} < \infty.$$

Now, the BorelCantelli lemma gives $\mathcal{L}(\boldsymbol{\theta}^{(k)}) \to 0$, almost surely.

$\square$

## B    Related Works: Further Details and Discussions

NUQLS shares notable similarities with, and exhibits distinct differences from, several prior works, which are discussed in-depth below.

### B.1    Linearized Laplace Approximation (LLA) Framework.

The popular LLA framework (Khan et al., 2019; Foong et al., 2019; Immer et al., 2021; Daxberger et al., 2021) shares a close connection with NUQLS, as both methods fundamentally rely on linearizing the network. However, a subtle yet significant distinction lies in their constructions: LLA begins by obtaining a proper distribution over the parameters and then draws parameter samples from it, while NUQLS bypasses this step and directly targets an approximation of the posterior distribution of the neural network.

As a direct consequence of this, in overparameterized settings, where the Hessian (or its Generalized Gauss-Newton approximation) is not positive definite, the LLA framework necessitates imposing an appropriate prior over the parameters to avoid degeneracy. In sharp contrast, NUQLS directly generates samples from the predictive distribution without introducing any artificial prior, thereby avoiding potential biases that such priors might impose on the covariance structure of the outputs and eliminating the need for additional hyperparameters. Consequently, as the LLA framework corresponds to a Bayesian generalized linear model (GLM), the weight-space vs. function-space duality in GLMs implies that its predictive distribution corresponds to a noisy GP with an NTK kernel. On the other hand, NUQLS leads to a noise-free GP. In interpolating regimes, where the model perfectly fits the data, the noise-free setting of NUQLS appears to be more suitable (Hodgkinson et al., 2023).

Another important consequence of this distinction arises in classification tasks. Due to the Laplace approximation, the LLA framework produces an independent GP for each output of the linearized model. In contrast, NUQLS captures the covariance between outputs, offering a more comprehensive representation of the predictive distribution.

Finally, for regression tasks, NUQLS offers additional flexibility by allowing the variance to be scaled post-hoc by a factor $\gamma$. This enables efficient hyperparameter tuning on a validation set without the need for retraining the model or optimizing a marginal likelihood–a level of flexibility not available in the LLA framework.

NUQLS can be seen as an extension of the "sample-then-optimize" framework for posterior sampling of large, finite-width NNs (Matthews et al., 2017). In this context, the work of Antorán et al. (2022), henceforth referred to as Sampling-LLA, enables drawing samples from the posterior distribution of the LLA in a manner analogous to Algorithm 1. In this approach, a series of regularized least-squares regression problems are constructed, and the collection of their solutions is shown to be distributed according to the LLA posterior. An EM algorithm is then employed for hyperparameter tuning. In addition to the fundamental differences between NUQLS and LLA-inspired methods mentioned earlier, Sampling-LLA has notable distinctions from Algorithm 1. First, the objective functions in Sampling-LLA have non-trivial minimum values. As a result, the convergence of SGD for such problems necessitates either a diminishing learning rate (Bubeck et al., 2015), which slows down convergence, or the adoption of variance reduction techniques (Roux et al., 2012; Shalev-Shwartz &

Zhang, 2013; Johnson & Zhang, 2013), which can introduce additional computational and memory overhead. By contrast, in overparameterized settings and under the assumptions of Theorem 3.2, the optimization problem in (3) allows interpolation. Consequently, SGD can employ a constant step size for convergence, improving optimization efficiency (Garrigos & Gower, 2023). Second, the inherent properties of the LLA framework, which require a positive definite Hessian or its approximation, necessitate regularizing the least-squares term in the subproblem of Sampling-LLA. This results in a strongly convex problem with a unique solution. Consequently, to generate a collection of solutions, Sampling-LLA constructs a random set of such subproblems, each involving fitting the linearized network to random outputs. These random outputs are sampled from a zero-mean Gaussian, with covariance given by the Hessian of the loss function, evaluated on the data. In contrast, the subproblem of NUQLS, i.e., (3), involves directly fitting the training data, and the ensemble of solutions is constructed as a result of random initialization of the optimization algorithm. Hence, the uncertainty captured by NUQLS arises naturally from the variance of solutions in the overparameterized regime, without the need for additional regularization or artificially constructed subproblems.

While the Sampling-LLA method enhances the scalability of LLA, the competing method Variational LLA (VaLLA) (Ortega et al., 2023) offers comparable or superior UQ performance while significantly reducing computation time. VaLLA achieves this by computing the LLA predictive distribution using a variational sparse GP with an NTK. Another competing LLA extension is Accelerated LLA (ELLA), which uses a Nyström approximation of the functional LLA covariance matrix (Deng et al., 2022), and seems to attain similar performance to VaLLA, again at a reduced cost compared to Sampling-LLA. We compare the performance of NUQLS to Sampling-LLA, VaLLA and ELLA in Appendix H.3.

Note that work by (Miani et al., 2024b) approximates an LLA by taking the covariance as the projection onto the null space of the GGN matrix, in order to compute a posterior that retains the same performance as the original network on training data. To compute samples, this work uses *alternating projections*.

## B.2 SNGP

We briefly describe the Spectral-Normalized Neural Gaussian Process (SNGP). SNGP lies at the intersection of feature-space methods and Bayesian methods, and similarly to NUQLS combines a GP with a NN. Specifically, SNGP adds a weight normalization step during training, and then replaces the output layer of NN with a GP that takes the feature extractor of the network as an input. While this method is also not post-hoc, and is not strictly a Bayesian method, we still compare SNGP with NUQLS in Figure 6 and Table 11, where we observe superior performance of NUQLS.

## B.3 Ensemble Framework.

In Lee et al. (2019), infinitely wide neural networks were shown to follow a GP distribution; however, this GP did not correspond to a true predictive distribution. Building on this, He et al. (2020) introduced a random, untrainable function to an infinitely wide neural network, deriving a GP predictive distribution using the infinite-width NTK. An ensemble of these modified NNs was then interpreted as samples from the GP's predictive posterior. In contrast, our method demonstrates that trained, finite-width, unmodified linearized networks are inherently samples from a GP with an NTK kernel. While their method, Bayesian Deep Ensembles (BDE), shares some conceptual similarities with ours, we omit it from our main experiments for several reasons. Firstly, the posterior analysis of BDE is valid only in the limit of large model sizes. For smaller datasets and models, the infinite-width NTK differs greatly from the empirical NTK (see Fort et al. (2020)). We can see that in this regime, in both Figure 4 and Table 8, that NUQLS outperforms BDE. Secondly, the method is computationally more expensive than DE, due to the need to compute the untrainable function ($\delta(.)$) and tune the scaling hyperparameter for classification. This contradicts our goal to provide a UQ method that is computationally more efficient than the state-of-the-art (SOTA) method DE while maintaining competitive performance. Finally, for larger model sizes, we provide comparison of NUQLS to BDE in Figure 6 and Table 11, where we observe that NUQLS outperforms BDE.

In Madras et al. (2019), the authors propose a method called "local ensembles", which perturbs the parameters of a trained network along directions of small loss curvature to create an ensemble of

*nearly* loss-invariant networks. The uncertainty of the original network is then quantified as the standard deviation across predictions in the ensemble. Building on this approach, Miani et al. (2024a), in a method called Sketched Lanczos Uncertainty (SLU), use the GGN approximation of the Hessian of the loss to identify these directions and introduce a sketched Lanczos algorithm (Meurant, 2006) to efficiently compute them. We compare the performance of NUQLS to SLU in Appendix H.2.

While similar, there are key differences between local ensembles and NUQLS. Local ensembles form a subspace of networks that attain *similar* loss values, allowing for directions with small but non-zero curvature, potentially encompassing more directions than those with exactly zero curvature. In contrast, when a solution to the linear optimization problem exists, NUQLS creates an ensemble of networks that all attain *exactly* the same loss. Additionally, NUQLS relies on first-order information to construct this ensemble, whereas local ensembles depend on second-order information.

While the zero-curvature directions of the GGN approximation correspond to the Jacobian's null space, the local ensembles method also includes directions with small but non-zero curvature. This inclusion introduces a notable distinction between the ensembles generated by local ensembles and those formed by NUQLS. Finally, while local ensembles employ low-rank approximations to compute directions efficiently, such approximations may inadequately represent the Hessian and fail to accurately capture the true curvature structure, as highlighted in Xie et al. (2022).

We also give mention to methods that seek to make DE more efficient, such as BatchEnsemble (BE) (Wen et al., 2020) and SnapshotEnsemble (Havasi et al., 2020). We note that these methods are yet to surpass DE as the SOTA in the literature. However,we compare NUQLS against BE in Figure 6 and Table 11, where we see that NUQLS shows superior performance, whilst remaining as scalable as BE. Further, BE requires modification to the structure of the network, and hence is not post-hoc.

### B.4   Neural Tangent Kernel Methods

The Procedural-Noise-Correcting (PNC) method of Huang et al. (2023) employs the limiting NTK for infinite-width networks to characterize the noise in the optimization process of training a neural network. It uses this to provide a frequentist confidence interval around test predictions of the neural network, and as it arises from the frequentist framework, PNC recovers statistical guarantees. In comparison, NUQLS does not rely on the limiting regime of the NTK, but rather employs the feature extraction of the empirical NTK to quantify uncertainty using a Bayesian framework. In Appendix H.1 we compare the performance of NUQLS vs. PNC. Unsurprisingly, given their constructions, we see that PNC outperforms NUQLS when the width of a network is extreme, while for finite-width networks that are fully trained, NUQLS outperforms PNC. This experiment illustrates that the methods are actually complementary, as they both excel in different settings.

## C   Uncertainty vs. Prediction

Here we discuss the use of common metrics to measure UQ ability in the BDL community. Generally, NLL, ECE and AUCROC (or the entropic version OODAUC) are used to measure the ability of a method to correctly quantify the uncertainty in a neural network. However, we argue that the use of these metrics for UQ is ill-advised, due to either a misalignment between the metric and the goal of UQ, or due to flawed measurements of uncertainty implicit in the metric. Specifically, we argue that variance over the softmax predictions *is* the uncertainty in the network, in the same way that the variance over probabilities *is* the uncertainty in a Dirichlet model. In contrast, NLL and AUCROC (OODAUC) are metrics for *prediction*, while ECE is a metric based on a flawed surrogate for uncertainty, calibration.

**Prediction**   While we will argue that uncertainty quantification *must* involve a computed variance term, and that common UQ metrics are either actually prediction metrics, or are based on a *prediction* of uncertainty, this discussion also begs the question: what about the purported benefits of BDL to prediction ability? As has been argued in Abe et al. (2022), it is often more economical to simply train a larger capacity model, than to use BDL to improve prediction quality. Hence, we restrict our main focus of NUQLS to its UQ ability. However, for completeness, we also evaluate the novel predictor of NUQLS; these results can be found in the Appendix G.

**Uncertainty**   Uncertainty can be divided into two categories: *aleatoric* uncertainty, or data uncertainty, and *epistemic* uncertainty, or model uncertainty. *Aleatoric* uncertainty arises from intrinsic randomness or noise in the data, and can often not be controlled. In contrast, *epistemic* uncertainty arises from having many potential models that may fit the data, and the uncertainty in not knowing which model is 'correct'. As this arises from a lack of knowledge of the original data-mapping, uncertainty in this manner can often be reduced through increased training data, using a larger model, training for longer etc. There are numerous benefits to untangling the aleatoric and epistemic uncertainty (Mukhoti et al., 2021; Huseljic et al., 2021; Chan et al., 2024). We can use the framework of a Dirichlet model, the conjugate prior of a Categorical distribution, to understand how to evaluate these uncertainties in the context of $K$-class classification. Assume there is some subspace $\mathcal{M}$ of the parameter space $\boldsymbol{\Theta} \in \mathbb{R}^p$, such that our network $\mathbf{f}(.; \boldsymbol{\theta}^*)$ attains 'good' test performance for any $\boldsymbol{\theta}^* \in \mathcal{M}$. If it is common across parameter values in $\mathcal{M}$ that $\mathbf{f}(\mathbf{x}; \boldsymbol{\theta}^*)$ has high entropy for some $\mathbf{x}$, i.e. $\mathbf{f}(\mathbf{x}; \boldsymbol{\theta}^*)$ outputs a prediction close to the uniform distribution with high probability across $\boldsymbol{\theta}^* \in \mathcal{M}$, then we are sure that the outcome is unpredictable in $\mathbf{x}$, i.e. that there is high aleatoric uncertainty. This equates to a Dirichlet model with parameters $\boldsymbol{\alpha} = (\alpha_1, \ldots, \alpha_K)$ where $\boldsymbol{\alpha}$ is uniform and of large magnitude. In contrast, assume that over all parameter values in $\mathcal{M}$, the softmax outputs of $\mathbf{f}(\mathbf{x}; \boldsymbol{\theta}^*)$ for a given $\mathbf{x}$ have very large variance. This means that no model can agree on the 'correct' data-mapping, and hence there is very large epistemic uncertainty. This is analogous to when the parameters $\boldsymbol{\alpha}$ of a Dirichlet model are small in magnitude. Our method, NUQLS, measures the variance over softmax outputs for high-performing parameters $\boldsymbol{\theta}^* \in \mathcal{M}$. Hence, we are measuring the epistemic uncertainty of the neural network.

**NLL**   Due to its probabilistic framework, NLL is often used as a metric for UQ ability. For $K$-class classification, up to a constant the NLL is just the cross-entropy loss

$$\text{NLL}(\boldsymbol{\theta}) = -\sum_{i=1}^n \sum_{k=1}^K y_{i,k} \log \mu_{i,k},$$

where $y_{i,k}$ corresponds to the $k$-th element of the one-hot encoding of the $i$-th label, and $\mu_{i,k} = \text{softmax}(\mathbf{f}(\mathbf{x}_i; \boldsymbol{\theta}))_k$. This loss is minimised as $\text{softmax}(\mathbf{f}(\mathbf{x}_i; \boldsymbol{\theta}))_k \to \mathbb{1}_{c_i}$, where $c_i$ is the correct label for input $\mathbf{x}_i$. While a lower NLL is generally considered to show better UQ ability, we can see that this metric is instead a smooth, probabilistic measurement of accuracy, where proximity of the predictor to the identity function is rewarded with lower NLL. *This metric does not measure uncertainty quantification ability.*

**ECE**   A common surrogate for uncertainty in neural networks is the predicted softmax probabilities. While neural networks generally output over-confident probabilities, many methods, such as temperature scaling and many Bayesian methods, attempt to create well-calibrated predictors. A well-calibrated output is one where the predicted softmax probabilities accurately convey the forecasted probability of seeing each of the $K$-classes. For example, of all test points where a network has a maximum predicted probability of $0.9$, we would hope that the network is $90\%$ accurate on these test points. Once a predictor is well-calibrated, the softmax probabilities are then seen to quantify uncertainty. However, there are issues with this:

1. Firstly, a point prediction from a network cannot be interpreted as the uncertainty in the network; uncertainty quantities must accompany a network prediction, for example with a variance term. At best, network confidence can be seen to *predict* the uncertainty.

2. Secondly, if softmax probability does in fact accurately predict the uncertainty, is this uncertainty due to data uncertainty, or model uncertainty? If a model accurately predicts that there is a $90\%$ chance that class $c$ is correct, does that uncertainty arise due to noise in the data, or some misspecification in the labeling process? Or is it due to the fact that the input is OoD, and thus the model is unsure of itself? Hence, we still require a variance term to ascertain what the epistemic uncertainty is.

3. Finally, calibration is only well-defined for ID test points. To see this, take a network trained on MNIST, and then tested on FashionMNIST. Any network prediction is meaningless on this test set, as the model has been trained for integer labels. The prediction that will minimize the calibration error will be the uniform prediction i.e. the prediction is $[0.1, \ldots, 0.1] \in \mathbb{R}^{10}$ for all points in the OoD test set. However, the accuracy on OoD test points *should be* $0$. Instead the predicted accuracy is $0.1$, which is what arises from randomly picking one class for each OoD test

point. Therefore, the predicted probabilites cannot be directly interpreted as forecasted probability of accuracy for OoD points.

In practice, calibration is tested using ECE. This metric bins predictions, and then computes the average confidence in each bin. This is compared with the average predicted accuracy, and an $l_2$ norm is taken over the differences. As was noted in Nixon et al. (2019), there are several issues with ECE. For examples, it is reductive in a multi-class setting, where the bottom $K-1$ probabilities do not contribute meaningfully to the error, and it suffers from the inherent sharpness of neural networks, where the majority of maximum probabilities tend to be very close to $1$, and thus this confidence range is highly over-represented. So we struggle to even tell when a method is well-calibrated.

This issue with ECE can be summarised as follows: confidence is a flawed *estimate* of the total uncertainty; further, it is hard to tell *when* a method is well-calibrated. It is thus evident that it is much better to *directly* model the uncertainty in the model, by calculating the variance over the softmax predictions. Once equipped with this model uncertainty, entropy (or network confidence) can be better trusted to inform of the aleatoric uncertainty.

**AUCROC**   The final metric commonly used for UQ is AUCROC (OODAUC). This metric uses the top softmax prediction (entropy) as a predictor, to attempt to detect wether a point is ID or OoD. While this is a very useful task, it is still a metric of *prediction*, as only the mean predictor is used to compute this value. As we have argued, UQ requires a variance term. Miani et al. (2024a) in fact use the variance over *logits* to compute the AUCROC score, and report this score for their method SLU in comparison to other leading UQ methods. In Appendix H.2 we show that in this metric, we outperform SLU.

**VMSP**   To combat the issues of the previous metrics, we compare the performance of Bayesian methods using the variance of the maximum softmax predictor, or VMSP. We first provide an explicit definition of VMSP: for a Bayesian method, we generally have a mean predictor $\mu : \mathbb{R}^d \to \mathbb{R}^c$ and a covariance function $\Sigma : \mathbb{R}^d \to \mathbb{R}^{c \times c}$, for example the mean and covariance of the linearized ensemble in the case of NUQLS, that output in the probit space. To compute VMSP for a given test point $\mathbf{x}^\star$, we first find $\hat{c} = \operatorname{argmax}_k \mu(\mathbf{x}^*)_k$, where $\mu(\mathbf{x}^\star)_k$ denotes the $k$-th output of $\mu(\mathbf{x}^\star)$. That is, we find the class that the Bayesian method predicts, given $\mathbf{x}^\star$. We then define VMSP $:= \Sigma(\mathbf{x}^\star)_{\hat{c},\hat{c}} = \sigma^2(\mathbf{x}^\star)_{\hat{c}}$, that is, the variance of this prediction.

We also introduce a pictorial method for evaluating the performance of Bayesian methods, using VMSP as the correct uncertainty measure. For a given dataset, we want a UQ method to provide low uncertainty for correctly predicted test points, and high uncertainty for incorrectly predicted or OOD test points. We then compare these distributions pictorially using a violin plot, and quantitatively using the median and skew values for the respective distributions. With the addition of a poorly-performing baseline model, we are able to easily compare the ability of UQ models to quantify uncertainty.

## D   Guarantees Against Mode Collapse

Given that all linearized models are initialized locally around the trained parameters, one may consider whether it is possible that all linear networks will converge to the same parameter solution, or mode of the linear loss. Fortunately, we are able to show that this will not occur, almost surely. To see this, we note that the solution to (3) for each linearized model is given by a unique row-space component (given our assumptions), plus a projection of the initialization $\mathbf{z}_i$ onto the null-space of the Jacobian. For mode-collapse, we would require the projection for two i.i.d initializations $\mathbf{z}_1, \mathbf{z}_2 \sim N(\mathbf{0}, \gamma^2 \mathbf{I})$ to be equal, or $(\mathbf{I} - \mathbf{J}_\mathcal{X}^\dagger \mathbf{J}_\mathcal{X})(\mathbf{z}_1 - \mathbf{z}_2) = \mathbf{0}$, where $\mathbf{z}_1 - \mathbf{z}_2 \neq \mathbf{0}$ with probability one. Hence, $\mathbf{z}_1 - \mathbf{z}_2 = \mathbf{J}_\mathcal{X}^\dagger \mathbf{J}_\mathcal{X}(\mathbf{z}_1 - \mathbf{z}_2)$, and thus $\mathbf{z}_1 - \mathbf{z}_2 \in \operatorname{Range}(\mathbf{J}_\mathcal{X}^T)$, i.e. $\mathbf{z}_1, \mathbf{z}_2 \in \operatorname{Range}(\mathbf{J}_\mathcal{X}^T)$. However, $\operatorname{Range}(\mathbf{J}_\mathcal{X}^T)$ is a low-dimensional subspace of $\mathbb{R}^p$, and as we are drawing i.i.d from a $p-$dimensional normal distribution (i.e. not degenerate), the probability of this event occurring is 0. Hence, we do not need to be concerned with all models converging to the same behaviour.

## E   Computational Cost

We now provide examination of the computational cost of NUQLS. We compare the computational complexity for an epoch of training for the neural network $f_\theta(x)$, for batch $x \in \mathbb{R}^{d \times n}$, and an

epoch of training for a single linear network $\hat{f}_\theta(x)$. We take approximations on the computational complexity of both forward-mode AD and backward-mode AD from Blondel & Roulet (2024, Chapter 8). Specifically, we take $[fp]$ as the computational complexity for evaluating $f_\theta(x)$. We note from Blondel & Roulet (2024, Chapter 8) that both a JVP and a VJP cost roughly $2 \times$ a forward pass in computational complexity and memory. Further, both a JVP and a JVP return a function evaluation. Now, a standard epoch of training for the neural network involves a forward-pass to compute the error, and then a backward-pass (VJP), hence the complexity is approx. $3[fp]$. The linearised network involves a JVP (which includes a function evaluation) to form the linear network, and a VJP to compute the gradient. Hence, the complexity for an epoch of training for the linearized network is approx. $4[fp]$. So we observe that each epoch for a linearized network is only $4/3 \times$ as expensive as for a neural network. In regards memory, we see that the memory requirement for both the linearized network and the neural network are similar. However, we generally train all linearized networks in parallel. For computational complexity, this will incur some additional cost, thought it will not be linear in number of networks, that is dependent upon the specific software and hardware. However, memory cost will scale linearly by number of networks, i.e. $2S \times M([fp])$, where $M([fp])$ is the memory cost for a forward-pass, and $S$ is the number of linear networks. As an example, we employ a batch size of 56 for ImageNet on ResNet50 with 10 ensemble members when using an 80GB H100 GPU, due to the large parameter count and large number of classes. Note that in the case where the training set is small, it is beneficial to compute and save in memory the Jacobian of the NN evaluated on the entire training set. We can then train the linear networks very quickly. This contributes to the impressive run-times seen in Tables 1 and 9.

## F    Hyper-parameter Tuning

NUQLS contains several hyper-parameters: the number of linear networks to be trained, the number of epochs and learning rate of training, and the variance of initialisation, $\gamma$. In this section, we discuss strategies to select optimal hyper-parameters.

### F.1    Regression

For regression, and with a sufficiently small learning rate, NUQLS samples from the distribution given in Remark 3.3. We can see that the variance of the predictions scales linearly with $\gamma^2$. Hence, we use the following framework to tune $\gamma$:

1. To obtain $\boldsymbol{\theta}_s$ in Algorithm 1, we initialise our parameters with a very small gamma, e.g. $\gamma = 0.01$. This enables (stochastic) gradient descent to converge quickly with a small learning rate.

2. We compute $\{\widetilde{\mathbf{f}}(\boldsymbol{\theta}_s^\star, \mathcal{X}_{\text{val}})\}_{s=1}^S$, where $\mathcal{X}_{\text{val}}$ is the inputs from a validation set. As per Remark 3.3, for each point $\mathbf{x} \in \mathcal{X}_{\text{val}}$, we compute the mean prediction as $\mu(x) = \text{SampleMean}\left(\{\widetilde{\mathbf{f}}(\boldsymbol{\theta}_s^\star, \mathbf{x})\}_{s=1}^S\right)$ and the variance as $\sigma_\gamma^2(x) = \text{SampleVariance}\left(\{\gamma\widetilde{\mathbf{f}}(\boldsymbol{\theta}_s^\star, \mathcal{X}_{\text{val}})\}_{s=1}^S\right)$ (note the scaling by $\gamma$ for only the variance). We then use these values to compute the ECE across the validation dataset $\mathcal{D}_{\text{val}}$.

3. As coverage of a confidence interval scales linearly with the size of the given standard deviation, and our computed standard deviation scales linearly with $\gamma$, we find that the ECE is convex in $\gamma$. Hence, we employ the Ternary search method (see Algorithm 2) to find the value $\hat{\gamma}$ that minimizes this validation ECE.

4. For a test point $\mathbf{x}^\star$, our mean prediction and variance is then $\mu(\mathbf{x}^\star)$ and $\sigma_{\hat{\gamma}}^2(\mathbf{x}^\star)$.

As can be seen in Table 1 and Table 9, this framework means that for regression our method is incredibly fast and computes well-calibrated variance values.

### F.2    Classification

For uncertainty quantification performance, as measured by VMSP in Figure 3, we find that as long as $\gamma$ is small, training SGD for a small amount of epochs generally gives small training loss, and hence provides good performance. This means that our method can be computed quite quickly in larger data/model settings.

---

**Algorithm 2** Ternary Search

---

$f$: function to minimize, $l$: left boundary of search space, $r$: right boundary of search space, $\delta$: tolerance, iter: iterations.

$i = 0$

**while** $|l - r| \geq \delta$ and $i <$iter **do**

   $l_{1/3} = l + (r - l)/3$

   $r_{1/3} = r - (r - l)/3$

   **if** $f(l_{1/3}) < f(r_{1/3})$ **then**

      $l = l_{1/3}$

   **else**

      $r = r_{1/3}$

   **end if**

**end while**

**return** $(l + r)/2$

---

Table 2: Image classification predictive performance, using LeNet5 on MNIST and FashionMNIST (FMNIST). Experiment was run 5 times with different random MAP initialisations to get standard deviation on metrics.

| Datasets | Method | NLL $\downarrow$ | ACC $\uparrow$ | ECE $\downarrow$ | OOD-AUC $\uparrow$ | AUC-ROC $\uparrow$ | Time (s) |
|---|---|---|---|---|---|---|---|
| | MAP | $0.034\pm_{0.002}$ | $\mathbf{0.990}\pm_{\mathbf{0.001}}$ | $0.008\pm_{0.001}$ | $0.888\pm_{0.008}$ | $0.886\pm_{0.008}$ | 257 |
| | NUQLS | $0.035\pm_{0.002}$ | $0.989\pm_{0.001}$ | $\mathbf{0.003}\pm_{\mathbf{0.000}}$ | $\mathbf{0.930}\pm_{\mathbf{0.026}}$ | $\mathbf{0.928}\pm_{\mathbf{0.026}}$ | 106 |
| **MNIST** | DE | $0.034\pm_{0.004}$ | $\mathbf{0.991}\pm_{\mathbf{0.000}}$ | $0.011\pm_{0.004}$ | $\mathbf{0.932}\pm_{\mathbf{0.009}}$ | $\mathbf{0.928}\pm_{\mathbf{0.009}}$ | 2845 |
| | MC-Dropout | $0.044\pm_{0.002}$ | $0.989\pm_{0.000}$ | $0.017\pm_{0.01}$ | $0.873\pm_{0.032}$ | $0.871\pm_{0.031}$ | 533 |
| | SWAG | $\mathbf{0.029}\pm_{\mathbf{0.003}}$ | $\mathbf{0.991}\pm_{\mathbf{0.000}}$ | $0.004\pm_{0.002}$ | $0.902\pm_{0.008}$ | $0.900\pm_{0.008}$ | 489 |
| | LLA* | $0.034\pm_{0.002}$ | $\mathbf{0.990}\pm_{\mathbf{0.001}}$ | $0.008\pm_{0.001}$ | $0.888\pm_{0.008}$ | $0.886\pm_{0.008}$ | 45 |
| | VaLLA | $0.034\pm_{0.002}$ | $\mathbf{0.990}\pm_{\mathbf{0.001}}$ | $0.008\pm_{0.001}$ | $0.889\pm_{0.008}$ | $0.886\pm_{0.008}$ | 1583 |
| | MAP | $0.298\pm_{0.007}$ | $0.891\pm_{0.3}$ | $\mathbf{0.006}\pm_{\mathbf{0.001}}$ | $0.840\pm_{0.022}$ | $0.804\pm_{0.021}$ | 158 |
| | NUQLS | $0.302\pm_{0.006}$ | $0.891\pm_{0.002}$ | $\mathbf{0.005}\pm_{\mathbf{0.002}}$ | $\mathbf{0.904}\pm_{\mathbf{0.007}}$ | $\mathbf{0.870}\pm_{\mathbf{0.006}}$ | 89 |
| **FMNIST** | DE | $0.288\pm_{0.002}$ | $0.896\pm_{0.001}$ | $0.013\pm_{0.001}$ | $0.876\pm_{0.003}$ | $0.836\pm_{0.003}$ | 1587 |
| | MC-Dropout | $0.306\pm_{0.007}$ | $0.892\pm_{0.003}$ | $0.026\pm_{0.002}$ | $0.856\pm_{0.021}$ | $0.813\pm_{0.019}$ | 291 |
| | SWAG | $\mathbf{0.283}\pm_{\mathbf{0.005}}$ | $\mathbf{0.899}\pm_{\mathbf{0.003}}$ | $0.018\pm_{0.002}$ | $0.817\pm_{0.023}$ | $0.783\pm_{0.022}$ | 264 |
| | LLA* | $0.298\pm_{0.007}$ | $0.891\pm_{0.003}$ | $\mathbf{0.006}\pm_{\mathbf{0.001}}$ | $0.841\pm_{0.022}$ | $0.805\pm_{0.021}$ | 26 |
| | VaLLA | $0.298\pm_{0.007}$ | $0.891\pm_{0.003}$ | $0.007\pm_{0.001}$ | $0.841\pm_{0.022}$ | $0.805\pm_{0.021}$ | 1583 |

# G   Evaluation of Predictive Mean

While the main focus of NUQLS is the variance term, to compute the epistemic uncertainty of a NN, we would also like to demonstrate the predictive ability of our novel predictive mean, for completeness.

## G.1   Image Classification - Predictive

We compare NUQLS against the MAP NN, DE, MC-Dropout, SWAG, LLA* and VaLLA, on the MNIST and FashionMNIST datasets, using the LeNet5 network. We compare test cross-entropy (NLL), test accuracy (ACC), ECE, and the AUC-ROC measurement for maximum softmax probability (AUC-ROC) and entropy (OOD-AUC) as the detector. The last two metrics evaluate a methods ability to detect out-of distribution points. We display the results in Table 2. We see that NUQLS performs the best in ECE, AUC-ROC and OOD-AUC, and is competitive in the other metrics, while having the second fastest wall-time.

# H   Further Experimental Results

## H.1   Comparison to PNC using Confidence Intervals

We use this section to compare the difference in performance of NUQLS vs PNC. To test the difference between the two methods, we tested NUQLS on the confidence intervals problem from Huang et al. (2023). In this experiment, an MLP with a single hidden-layer is trained on $n$ datapoints

Table 3: Evalaution of coverage (CR) and width (IW) of computed confidence intervals from toy problem, for a neural network with **extreme** width and **partial** training. A computed coverage exceeding the expected coverage is bolded. The mean prediction (MP) is also provided.

| | PNC | | | NUQLS | | |
|---|---|---|---|---|---|---|
| | 95%CI (CR/IW) | 90%CI (CR/IW) | MP | 95%CI (CR/IW) | 90%CI (CR/IW) | MP |
| $d=2$, $n=128$ | **0.98**/0.0437 | **0.95**/0.0323 | 0.1998 | 0.93/0.0357 | **0.92**/0.0299 | 0.2047 |
| $d=4$, $n=256$ | **0.98**/0.0411 | **0.95**/0.0304 | 0.3991 | 0.92/0.0596 | 0.86/0.0500 | 0.4084 |

Table 4: Evalaution of coverage (CR) and width (IW) of computed confidence intervals from toy problem, for a neural network with **finite** width and **full** training. A computed coverage exceeding the expected coverage is bolded. The mean prediction (MP) is also provided.

| | PNC | | | NUQLS | | |
|---|---|---|---|---|---|---|
| | 95%CI (CR/IW) | 90%CI (CR/IW) | MP | 95%CI (CR/IW) | 90%CI (CR/IW) | MP |
| $d=2$, $n=128$ | 0.8/0.136 | 0.72/0.0105 | 0.2022 | **0.99**/0.0135 | **0.96**/0.0114 | 0.2012 |
| $d=4$, $n=256$ | **0.96**/0.0437 | **0.90**/0.0336 | 0.4045 | **0.97**/0.0313 | **0.97**/0.0313 | 0.4030 |
| $d=8$, $n=512$ | 0.88/0.0740 | 0.88/0.0568 | 0.8078 | **1.00**/0.0667 | **0.98**/0.0559 | 0.8050 |
| $d=16$, $n=128$ | 0.8/0.1443 | 0.8/0.1108 | 1.6265 | **1.00**/0.1350 | **0.98**/0.1133 | 1.6121 |

from $U([0,0.2]^d)$, where the target is $y = \sum_{i=1}^{d} sin(x_i) + \epsilon$, and $\epsilon$ is a small noise term. The method is then asked to form a confidence interval around the prediction for $\mathbf{x} = (0.1, 0.1, \ldots, 0.1)$. This setup is repeated several times, i.e. the network and training data are randomly initialized, and the coverage and average width of the confidence intervals are recorded, as well as the mean prediction. For more details, we refer the reader to Huang et al. (2023). When the width of the network is $32 \times n$, and the network is only partially trained with 80 epochs and a learning rate of 0.01, NUQLS performs poorly compared to PNC, as can be seen in Table 3. Note that bolded numbers indicate intervals that have reached or exceeded the expected coverage. Further, smaller width intervals are preferred.

We note that scaling the width of a network by $32 \times n$ is rare in practice, and that such a wide network is difficult to train. If we instead form an MLP with width equal to the number of training points, and increase epochs to 100 and learning rate to 0.5, so that the network is properly trained, NUQLS far outperforms PNC, as can be seen in Table 4. We note that for these experiments we tuned the $\gamma$ hyper-parameter for NUQLS on a small validation set. We conclude that NUQLS and PNC are in fact complementary methods. For infinite-width networks near initialization, PNC performs well while NUQLS struggles. Conversely, for finite-width networks trained to a minimum, NUQLS excels while PNC performs poorly. We believe the latter regime is more representative of practical scenarios, where NUQLS offers significantly better performance.

## H.2 Comparison to SLU

Here we compare the performance of our method against the competing SLU method. See Appendix B for a discussion of the differences between NUQLS and SLU. Due to the extensive experimental details given in (Miani et al., 2024a), we run certain experiments from this work and report the performance of NUQLS against the SLU results found in (Miani et al., 2024a). In Table 5, we compare NUQLS against SLU on OoD detection using the AUC-ROC metric, on a smaller single-layer MLP and a larger LeNet model. The MLP is trained on the MNIST dataset, while the LeNet model is trained on the FashionMNIST dataset. For MNIST, the OoD datasets are FashionMNIST, KMNIST, and a Rotated MNIST dataset, where for each experiment run, we compute the average AUC-ROC score over a range of rotation angles $(15, 30, 45, 60, 75, 90, 105, 120, 135, 150, 165, 180)$. For FashionMNIST, the OoD datasets are MNIST, and the average Rotated FashionMNIST dataset. The AUC-ROC metric was calculated using the **variance of the logits**, summed over the classes for each test point, as a score. We observe that NUQLS has a better AUC-ROC value over all OoD datasets, often by a significant margin.

## H.3 Comparison to Sampling-LLA, VaLLA and ELLA

We now compare NUQLS against Sampling-LLA, VaLLA and ELLA. Due to issues with convergence, performance, memory usage and package compatibility when either running source

Table 5: Comparing performance of NUQLS against SLU method. Metric given is the AUC-ROC, computed using the variance of the logits, summed over the classes for each test point, as a score. AUC-ROC measures ability of a method to differentiate between ID and OoD points. We see that NUQLS out-performs SLU in these experiments.

| Model | **MLP** $p = 15k$ | | | **LeNet** $p = 40k$ | |
| ID Data | **MNIST vs** | | | **FashionMNIST** | |
| OoD Data | **FashionMNIST** | **KMNIST** | **Rotation (avg)** | **MNIST** | **Rotation (avg)** |
|---|---|---|---|---|---|
| SLU | $0.26\pm_{0.02}$ | $0.42\pm_{0.04}$ | $0.59\pm_{0.02}$ | $\mathbf{0.94}\pm_{\mathbf{0.01}}$ | $0.74\pm_{0.03}$ |
| NUQLS | $\mathbf{0.67}\pm_{\mathbf{0.07}}$ | $\mathbf{0.79}\pm_{\mathbf{0.02}}$ | $\mathbf{0.74}\pm_{\mathbf{0.01}}$ | $\mathbf{0.95}\pm_{\mathbf{0.02}}$ | $\mathbf{0.91}\pm_{\mathbf{0.01}}$ |

Table 6: Comparing performance of NUQLS against Sampling-LLA, VaLLA and ELLA on MNIST, trained used a 2-layer MLP with 200 hidden units in each layer, and $\mathrm{tanh}$ activation. Original results taken from (Ortega et al., 2023).

| Model | ACC | NLL | ECE | BRIER | OOD-AUC |
|---|---|---|---|---|---|
| ELLA | 97.6 | 0.076 | 0.008 | 0.036 | 0.905 |
| Sampled LLA | 97.6 | 0.087 | 0.026 | 0.040 | **0.954** |
| VaLLA 100 | 97.7 | 0.076 | 0.010 | 0.036 | 0.916 |
| VaLLA 200 | 97.7 | 0.075 | 0.010 | 0.035 | 0.921 |
| NUQLS | $\mathbf{98.0}\pm_{\mathbf{0.1}}$ | $\mathbf{0.065}\pm_{\mathbf{0.003}}$ | $\mathbf{0.005}\pm_{\mathbf{0.001}}$ | $\mathbf{0.031}\pm_{\mathbf{0.001}}$ | $\mathbf{0.953}\pm_{\mathbf{0.006}}$ |

code or implementing methods from instructions given in (Antorán et al., 2022) and (Ortega et al., 2023), we instead compare NUQLS against the results for Sampling-LLA, VaLLA and ELLA taken verbatim from (Ortega et al., 2023, Figure 3). We train a 2-layer MLP, with 200 hidden-units in each layer, on the MNIST dataset, according to the experimental details given in (Ortega et al., 2023). We then compare the accuracy, NLL, ECE, Brier score and the OOD-AUC metric of NUQLS against those reported for Sampling-LLA, VaLLA and ELLA. The results are shown in Table 6. We display the mean and standard deviation for NUQLS; as is quoted in (Ortega et al., 2023), the standard deviation for the other methods was below $10^{-4}$ in magnitude, and was thus omitted. We see that NUQLS outperforms Sampling-LLA, VaLLA and ELLA in accuracy, NLL, ECE, and Brier score, and is within one-sixth of a standard deviation of the leading OOD-AUC value, attained by Sampling-LLA. While we cannot directly comment on differences in computation time between our method and these LLA extensions, we were able to run the source code for VaLLA for the experiments in Table 2, where NUQLS was an order-of-magnitude faster than VaLLA in wall-time. In (Ortega et al., 2023, Figure 3 (right)), we also observe that VaLLA is an order-of-magnitude faster than both Sampling-LLA and ELLA. We also compare with (Ortega et al., 2023, Table 1), where a ResNet20 and ResNet34 model is trained on CIFAR-10. The results are displayed in Table 7. We see that NUQLS is competitive with all other methods in this setting.

Table 7: Comparing performance of NUQLS against Sampling-LLA, VaLLA and ELLA on CIFAR-10, with ResNet20 and ResNet32. Original results taken from (Ortega et al., 2023). Purple figures correspond to the top result, while blue figures are the second-best result.

| | ResNet20 | | | ResNet32 | | |
| Model | ACC | NLL | ECE | ACC | NLL | ECE |
|---|---|---|---|---|---|---|
| ELLA | 92.5 | 0.233 | 0.009 | 93.5 | 0.215 | 0.008 |
| Sampled LLA | 92.5 | 0.231 | 0.006 | 93.5 | 0.217 | 0.008 |
| VaLLA | 92.6 | 0.228 | 0.007 | 93.5 | 0.211 | 0.007 |
| NUQLS | 92.5 | 0.228 | 0.006 | 93.4 | 0.215 | 0.007 |

Table 8: Comparing performance of NUQLS and BDE on UCI regression tasks. We see that NUQLS outperforms BDE on all tasks.

| Dataset | Method | RMSE ↓ | NLL ↓ | ECE ↓ |
|---------|--------|--------|-------|-------|
| **Energy** | BDE | $0.416\pm_{0.039}$ | $-0.125\pm_{0.212}$ | $0.008\pm_{0.005}$ |
| | NUQLS | $\mathbf{0.047}\pm_{\mathbf{0.006}}$ | $\mathbf{-2.400}\pm_{\mathbf{0.209}}$ | $\mathbf{0.002}\pm_{\mathbf{0.002}}$ |
| **Concrete** | BDE | $0.714\pm_{0.054}$ | $1.563\pm_{0.449}$ | $0.063\pm_{0.012}$ |
| | NUQLS | $\mathbf{0.330}\pm_{\mathbf{0.047}}$ | $\mathbf{-0.316}\pm_{\mathbf{0.501}}$ | $\mathbf{0.003}\pm_{\mathbf{0.001}}$ |
| **Kin8nm** | BDE | $0.851\pm_{0.037}$ | $1.383\pm_{0.582}$ | $0.042\pm_{0.012}$ |
| | NUQLS | $\mathbf{0.252}\pm_{\mathbf{0.005}}$ | $\mathbf{-0.796}\pm_{\mathbf{0.025}}$ | $\mathbf{0.000}\pm_{\mathbf{0.000}}$ |

## H.4 Comparison to BDE

Figure 4 displays the performance of BDE on the toy regression problem from Figure 1. We also compare BDE against NUQLS on several UCI regression tasks in Table 8.

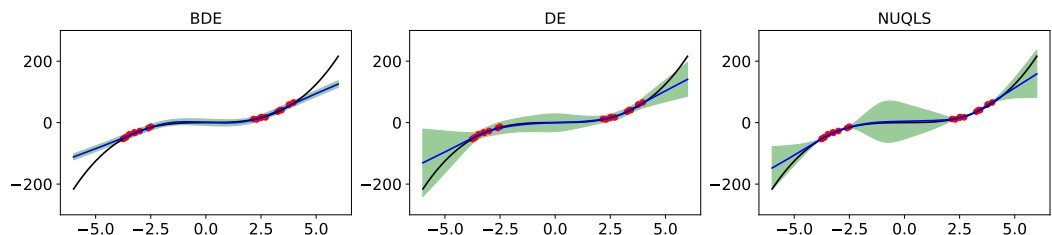

Figure 4: Comparison of BDE, DE and NUQLS on the toy regression problem from Figure 1. We can see that the uncertainty of the BDE method is quite small.

## H.5 UCI Regression

We present the results for select UCI regression datasets in Table 1 in the main body; we present results for several more datasets in Table 9.

## H.6 eNUQLS

In Figure 5 we demonstrate the performance of an ensembled version of NUQLS, eNUQLS. This method is similar to a Mixture of Laplace Approximations (Eschenhagen et al., 2021). We observe excellent separation of the variances between correct and incorrect/OoD test groups for eNUQLS, especially for CIFAR-10 on ResNet9. Note that there is significant cost to ensembling our method, and we provide this figure simply to illustrate performance capacity.

## H.7 Additional VMSP Results

We use this section to expand the empirical evaluation of NUQLS. In the top left of Figure 6, we observe that on a ResNet50 model trained on SVHN, with CIFAR-10 as the OoD test data, NUQLS outperforms all other methods. In the top right of Figure 6, we evaluated NUQLS on the ImageNet dataset in order to display the scalability of our method to larger datasets. We employed the pre-trained weights for a ResNet50, as found on *torch.hub*, and used the ImageNet-o dataset as our OOD test set (Hendrycks et al., 2021). Due to resource budget constraints, we were only able to compare NUQLS against a baseline method, though future research could include a comparison of other methods against NUQLS, using VMSP, on ImageNet. As can be seen, we see excellent separation between correct and incorrect predictions, while only adequate separation between correct and OOD points. We note from Bitterwolf et al. (2023) that approximately 21% of the images in ImageNet-o are actually in-distribution; hence, we see that NUQLS correctly quantifies the variance of these test sets. We also provide comparison of NUQLS with additional competing methods. In the bottom left of Figure 6, we compare NUQLS against the BE method. Note that BE requires modification to the structure of a neural network; hence, we compare BE on a modified WideResNet-34-1, of which an implementation of the correct modifications existed (Franchi et al., 2023), trained on FashionMNIST.

Table 9: Comparing performance of NUQLS, DE, LLA and SWAG on UCI regression tasks. NUQLS performs as well as or better than all other methods, while showing a speed up over other methods; this speed up increases with the size of the datasets. Note that for the Song dataset, the LLA method uses a diagonal covariance structure due to memory constraints: this is denoted as LLA-D.

| Dataset | Method | RMSE ↓ | NLL ↓ | ECE ↓ | Time(s) |
|---------|--------|--------|-------|-------|---------|
| **Concrete** | NUQLS | $\mathbf{0.330}\pm_{0.047}$ | $-0.316\pm_{0.501}$ | $\mathbf{0.003}\pm_{0.001}$ | $\mathbf{7.339}\ (\mathbf{0.185})$ |
| | DE | $0.379\pm_{0.019}$ | $-0.574\pm_{0.098}$ | $0.002\pm_{0.002}$ | $29.047$ |
| | LLA | $\mathbf{0.333}\pm_{0.050}$ | $-0.294\pm_{0.479}$ | $0.003\pm_{0.002}$ | $7.451(0.297)$ |
| | SWAG | $0.334\pm_{0.050}$ | $-0.562\pm_{0.224}$ | $0.009\pm_{0.006}$ | $43.416\ (36.262)$ |
| **Naval** | NUQLS | $\mathbf{0.049}\pm_{0.012}$ | $\mathbf{-2.546}\pm_{0.134}$ | $\mathbf{0.002}\pm_{0.002}$ | $\mathbf{11.360}\ (\mathbf{0.295})$ |
| | DE | $0.076\pm_{0.006}$ | $-1.761\pm_{0.250}$ | $0.093\pm_{0.040}$ | $96.570$ |
| | LLA | $0.070\pm_{0.022}$ | $25.292\pm_{17.570}$ | $0.192\pm_{0.029}$ | $140.724\ (129.659)$ |
| | SWAG | $1.130\pm_{1.500}$ | $0.303\pm_{1.091}$ | $0.084\pm_{0.022}$ | $103.727\ (92.662)$ |
| **CCPP** | NUQLS | $0.244\pm_{0.008}$ | $-0.885\pm_{0.020}$ | $\mathbf{0.000}\pm_{0.000}$ | $\mathbf{6.698}\ (\mathbf{0.174})$ |
| | DE | $\mathbf{0.227}\pm_{0.006}$ | $\mathbf{-1.009}\pm_{0.041}$ | $0.002\pm_{0.003}$ | $79.791$ |
| | LLA | $0.243\pm_{0.007}$ | $29.420\pm_{4.565}$ | $0.163\pm_{0.008}$ | $38.572\ (32.048)$ |
| | SWAG | $0.252\pm_{0.012}$ | $-0.849\pm_{0.038}$ | $0.001\pm_{0.002}$ | $73.357\ (66.833)$ |
| **Wine** | NUQLS | $\mathbf{0.789}\pm_{0.042}$ | $\mathbf{0.284}\pm_{0.066}$ | $\mathbf{0.001}\pm_{0.000}$ | $\mathbf{1.164}\ (\mathbf{0.115})$ |
| | DE | $\mathbf{0.789}\pm_{0.041}$ | $0.320\pm_{0.109}$ | $\mathbf{0.001}\pm_{0.001}$ | $13.241$ |
| | LLA | $0.792\pm_{0.041}$ | $1.012\pm_{0.182}$ | $0.009\pm_{0.004}$ | $1.389\ (0.340)$ |
| | SWAG | $0.798\pm_{0.038}$ | $0.367\pm_{0.103}$ | $0.005\pm_{0.003}$ | $12.856\ (11.807)$ |
| **Yacht** | NUQLS | $\mathbf{0.042}\pm_{0.013}$ | $-1.561\pm_{2.319}$ | $0.012\pm_{0.010}$ | $\mathbf{3.390}\ (\mathbf{0.164})$ |
| | DE | $0.647\pm_{0.121}$ | $-2.032\pm_{0.349}$ | $0.016\pm_{0.008}$ | $40.132$ |
| | LLA | $0.043\pm_{0.014}$ | $\mathbf{-2.733}\pm_{0.468}$ | $\mathbf{0.011}\pm_{0.006}$ | $3.403\ (0.177)$ |
| | SWAG | $0.044\pm_{0.014}$ | $-2.565\pm_{0.118}$ | $0.067\pm_{0.025}$ | $19.408\ (15.822)$ |
| **Song** | NUQLS | $\mathbf{0.839}\pm_{0.014}$ | $0.646\pm_{0.056}$ | $0.001\pm_{0.000}$ | $\mathbf{295.058}\ (\mathbf{91.673})$ |
| | DE | $0.846\pm_{0.006}$ | $\mathbf{0.180}\pm_{0.013}$ | $0.005\pm_{0.000}$ | $2562.789$ |
| | LLA-D | $0.851\pm_{0.029}$ | $0.456\pm_{0.093}$ | $\mathbf{0.000}\pm_{0.000}$ | $413.814\ (210.429)$ |
| | SWAG | $0.845\pm_{0.002}$ | $0.680\pm_{0.062}$ | $0.003\pm_{0.001}$ | $3477.825\ (3274.440)$ |

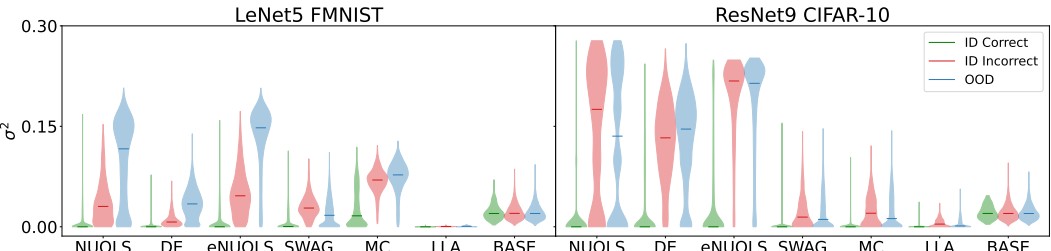

Figure 5: Violin plot of VMSP, with an ensembled version of NUQLS, eNUQLS, included.

We again observe that NUQLS shows greater separation than BE for VMSP across correct predictions versus incorrect or OOD predictions. Finally, in the bottom right of Figure 6, we compare NUQLS against further competing methods, specifically BDE, SGLD (Welling & Teh, 2011) (which we include to demonstrate the inferior performance of MCMC methods in this regime) and SNGP on ResNet9 trained on FMNIST (note SNGP requires modifications to the training procedure). We see that BDE performs similarly to DE, and SLGD performs similarly to SWAG, which is unsurprising considering their respective constructions. We also observe that NUQLS still performs the strongest across all methods using the VMSP violin plots. The respective median and skew values for Figure 6 are displayed in Table 10 and Table 11; these values also evidence the strong performance of NUQLS.

## H.8   Image Classification Skew and Median

The median and sample skew for the VMSP in Figure 3 and Figure 6 is found in Table 10 and Table 11.

Table 10: Sample median and sample skew of variance from Figure 3 and Figure 6. IDC = ID correct, IDIC = ID incorrect, FMNIST = FashionMNIST. The median and skew for a method is compared against a baseline method. We expect positive skew for the ID correct (IDC) variances, and a negative skew for the ID incorrect (IDIC) and OoD variances. If the median or skew for a method is worse than the corresponding baseline, then it is written in gray.

| **Median** | | NUQLS | DE | SWAG | MC | LLA | BASE |
|---|---|---|---|---|---|---|---|
| **ResNet9** | **IDC** $\downarrow$ | $4.76 \times 10^{-15}$ | $1.31 \times 10^{-5}$ | $4.64 \times 10^{-7}$ | $5.05 \times 10^{-7}$ | $9.02 \times 10^{-10}$ | 0.020 |
| **FMNIST** | **IDIC** $\uparrow$ | 0.182 | 0.040 | 0.048 | 0.004 | 0.004 | 0.020 |
| | **OoD** $\uparrow$ | 0.217 | 0.109 | 0.075 | 0.004 | 0.009 | 0.020 |
| **ResNet50** | **IDC** $\downarrow$ | 0.00 | $8.29 \times 10^{-8}$ | 0.018 | $1.49 \times 10^{-9}$ | $1.70 \times 10^{-7}$ | 0.019 |
| **CIFAR-10** | **IDIC** $\uparrow$ | 0.178 | 0.120 | 0.123 | $1.32 \times 10^{-4}$ | 0.008 | 0.020 |
| | **OoD** $\uparrow$ | 0.178 | 0.106 | 0.134 | $6.74 \times 10^{-5}$ | 0.004 | 0.020 |
| **ResNet50** | **IDC** $\downarrow$ | $1.00 \times 10^{-8}$ | $8.73 \times 10^{-4}$ | 0.0449 | $1.12 \times 10^{-4}$ | $3.25 \times 10^{-4}$ | 0.020 |
| **CIFAR-100** | **IDIC** $\uparrow$ | 0.211 | 0.0624 | 0.101 | $5.93 \times 10^{-3}$ | 0.0225 | 0.020 |
| | **OoD** $\uparrow$ | 0.214 | 0.0665 | 0.0956 | $5.46 \times 10^{-3}$ | 0.0199 | 0.020 |
| **ResNet50** | **IDC** $\downarrow$ | $8.61 \times 10^{-15}$ | $3.59 \times 10^{-6}$ | $1.71 \times 10^{-6}$ | $2.18 \times 10^{-8}$ | $1.61 \times 10^{-6}$ | 0.020 |
| **SVHN** | **IDIC** $\uparrow$ | 0.217 | 0.0569 | 0.0302 | $3.96 \times 10^{-4}$ | 0.0241 | 0.020 |
| | **OoD** $\uparrow$ | 0.233 | 0.127 | 0.0789 | $7.51 \times 10^{-4}$ | 0.0252 | 0.020 |
| **ResNet50** | **IDC** $\downarrow$ | 0.00 | - | - | - | - | 0.020 |
| **ImageNet** | **IDIC** $\uparrow$ | 0.232 | - | - | - | - | 0.020 |
| | **OoD** $\uparrow$ | 0.100 | - | - | - | - | 0.020 |
| | | | | | | | |
| **Sample Skew** | | | | | | | |
| **ResNet9** | **IDC** $\uparrow$ | 2.4 | 3.51 | 3.72 | 4.92 | 6.46 | 1.01 |
| **FMNIST** | **IDIC** $\downarrow$ | $-0.615$ | 0.928 | 0.378 | 1.60 | 1.18 | 1.11 |
| | **OoD** $\downarrow$ | $-1.7$ | 0.321 | 0.076 | 1.69 | 0.926 | 1.11 |
| **ResNet50** | **IDC** $\uparrow$ | 2.97 | 2.68 | 4.02 | 4.55 | 5.14 | 1.14 |
| **CIFAR-10** | **IDIC** $\downarrow$ | $-0.15$ | $-0.05$ | 0.66 | 2.45 | 0.816 | 1.09 |
| | **OoD** $\downarrow$ | 0.02 | $-0.01$ | 0.77 | 2.72 | 0.96 | 1.05 |
| **ResNet50** | **IDC** $\uparrow$ | 1.08 | 1.48 | 0.508 | 2.87 | 1.8 | 1.07 |
| **CIFAR-100** | **IDIC** $\downarrow$ | $-1.15$ | 0.241 | $-0.47$ | 2.05 | 0.228 | 1.09 |
| | **OoD** $\downarrow$ | $-1.37$ | 0.269 | $-0.233$ | 1.85 | 0.217 | 1.13 |
| **ResNet50** | **IDC** $\uparrow$ | 2.2 | 2.42 | 3.52 | 6.91 | 4.15 | 0.989 |
| **SVHN** | **IDIC** $\downarrow$ | $-1.31$ | 0.52 | 0.878 | 1.64 | 0.601 | 1.12 |
| | **OoD** $\downarrow$ | $-1.78$ | $-0.47$ | $-0.0119$ | 1.18 | 0.296 | 1.06 |
| **ResNet50** | **IDC** $\uparrow$ | 1.52 | - | - | - | - | 1.21 |
| **ImageNet** | **IDIC** $\downarrow$ | $-0.88$ | - | - | - | - | 1.10 |
| | **OoD** $\downarrow$ | 0.179 | - | - | - | - | 0.985 |

Table 11: Sample median and sample skew of variance from Figure 6. IDC = ID correct, IDIC = ID incorrect, FMNIST = FashionMNIST. The median and skew for a method is compared against a baseline method. We expect positive skew for the ID correct (IDC) variances, and a negative skew for the ID incorrect (IDIC) and OoD variances. If the median or skew for a method is worse than the corresponding baseline, then it is written in gray.

| **Median** | | NUQLS | BE | BDE | SGLD | SNGP | BASE |
|---|---|---|---|---|---|---|---|
| **WRN-34-1** | **IDC** $\downarrow$ | 0.00 | $8.31 \times 10^{-5}$ | - | - | - | 0.020 |
| **FMNIST** | **IDIC** $\uparrow$ | 0.178 | 0.045 | - | - | - | 0.020 |
| | **OoD** $\uparrow$ | 0.173 | 0.107 | - | - | - | 0.020 |
| **ResNet9** | **IDC** $\downarrow$ | $4.76 \times 10^{-15}$ | - | $3.35 \times 10^{-20}$ | $6.79 \times 10^{-8}$ | $9.19 \times 10^{-7}$ | 0.020 |
| **FMNIST** | **IDIC** $\uparrow$ | 0.182 | - | 0.00915 | 0.023 | 0.000322 | 0.020 |
| | **OoD** $\uparrow$ | 0.217 | - | 0.0504 | 0.0352 | 0.000342 | 0.020 |
| | | | | | | | |
| **Sample Skew** | | | | | | | |
| **WRN-34-1** | **IDC** $\uparrow$ | 3.1 | 2.8 | - | - | - | 1.01 |
| **FMNIST** | **IDIC** $\downarrow$ | $-0.342$ | 0.944 | - | - | - | 1.11 |
| | **OoD** $\downarrow$ | $-0.0875$ | $-0.272$ | - | - | - | 1.11 |
| **ResNet9** | **IDC** $\uparrow$ | 2.4 | - | 6.46 | 5.49 | 29.8 | 0.93 |
| **FMNIST** | **IDIC** $\downarrow$ | $-0.615$ | - | 1.43 | 1.23 | 24.4 | 1.15 |
| | **OoD** $\downarrow$ | $-1.7$ | - | 0.515 | 1.1 | 1.21 | 1.05 |

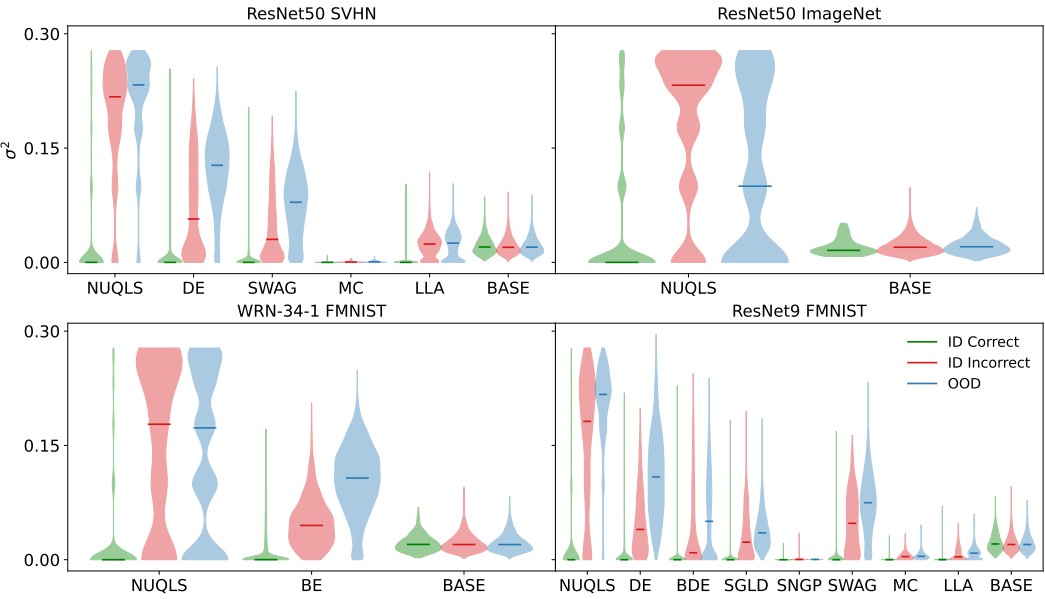

Figure 6: Violin plot of VMSP, for (top left) ResNet50 SVHN, (top right) ResNet50 ImageNet, (bottom left) WRN-34-1 FMNIST, and (bottom right) ResNet9 FMNIST.

# I  Experiment Details

All experiments were run either on an Intel i7-12700 CPU (toy regression), or on an H100 80GB GPU (UCI regression and image classification). Where multiple experiments were run, mean and standard deviation were presented.

## I.1  NTK Convergence

For this experiment, 2 randomly sampled sets of 100 Gaussian inputs of dimension 5, and 100 scalar Gaussian targets, were used as training points and test points respectively. The weights of the MLP were initialised to $\mathcal{N}(0, 1)$; no bias terms were used. To train, GD was employed with a Nesterov momentum parameter of $0.9$, a learning rate of $0.1$, and $5000$ epochs. For NUQLS, a learning rate of $0.1$ was used, and $\gamma$ was set to $1$. Error bars on Figure 2 are the $95\%$ sample confidence interval.

## I.2  Toy Regression

We use a 1-layer MLP, with a width of 50 and SiLU activation. For the maximum a posteriori (MAP) network, we train for 10000 epochs, with a learning rate of $0.001$, using the Adam optimizer and the PyTorch polynomial learning rate scheduler, with parameters total_iters = epochs, power = $0.5$. For DE, each network in the ensemble outputs a heteroskedastic variance, and is trained using a Gaussian NLL, with 2000 epochs and a learning rate of $0.05$. We combine the predictions of the ensembles as per (Lakshminarayanan et al., 2017). Both DE and NUQLS use 10 realizations. The $\gamma$ hyper-parameter in NUQLS is set to 5, and each linear realization is trained for 1000 epochs with a learning rate of $0.001$, using SGD with a momentum parameter of $0.9$. In SWAG, the MAP network is trained for a further 10000 epochs, using the same learning rate, and the covariance is formed with a rank-10 approximation. A prior precision of $0.1$ and $1$ is used for LLA and LA respectively, as well as the full covariance matrix. The variational inference method used is Bayes By Backprop (Blundell et al., 2015), as deployed in the Bayesian Torch package (Krishnan et al., 2022). The prior parameters are ($\mu = 0, \sigma = 1$), and the posterior is initialized at ($\mu = 0, \rho = -3$). For SWAG, LLA, LA and VI, 1000 MC sample were taken at test time. These design choices gave the best performance for this problem.

Table 12: Training procedure for UCI regression results in Table 1 and Table 9.

| NN | Energy | Concrete | Kin8nm | Naval | CCPP | Wine | Yacht | Protein | Song |
|---|---|---|---|---|---|---|---|---|---|
| Learning Rate | $10^{-2}$ | $10^{-2}$ | $10^{-2}$ | $10^{-2}$ | $10^{-2}$ | $10^{-2}$ | $10^{-2}$ | $10^{-2}$ | $10^{-2}$ |
| Epochs | 1500 | 1000 | 500 | 150 | 100 | 100 | 1000 | 250 | 50 |
| Weight Decay | $10^{-5}$ | $10^{-5}$ | $10^{-5}$ | $10^{-4}$ | $10^{-5}$ | $10^{-4}$ | $10^{-5}$ | $10^{-4}$ | 0 |
| Optimizer | Adam | Adam | SGD | SGD | Adam | SGD | Adam | SGD | SGD |
| Scheduler | PolyLR | PolyLR | None | None | PolyLR | None | PolyLR | None | None |
| MLP Size | $[150]$ | $[150]$ | $[100, 100]$ | $[150, 150]$ | $[100, 100]$ | $[100]$ | $[100]$ | $[150, 200, 150]$ | $[1000, 1000, 500, 50]$ |
| **NUQLS** | | | | | | | | | |
| Learning Rate | $10^{-2}$ | $10^{-2}$ | $10^{-2}$ | $10^{-2}$ | $10^{-2}$ | $10^{-2}$ | $10^{-2}$ | $10^{-2}$ | $10^{-3}$ |
| Epochs | 150 | 100 | 50 | 15 | 10 | 10 | 100 | 25 | 10 |
| **DE** | | | | | | | | | |
| Learning Rate | $10^{-3}$ | $10^{-3}$ | $10^{-2}$ | $10^{-3}$ | $10^{-2}$ | $10^{-2}$ | $10^{-2}$ | $10^{-2}$ | $10^{-2}$ |
| Epochs | 1500 | 300 | 100 | 100 | 100 | 100 | 1000 | 250 | 50 |
| Optimizer | Adam | Adam | Adam | Adam | Adam | Adam | Adam | Adam | Adam |
| Scheduler | None | None | Cosine | None | None | None | Cosine | None | None |
| **Experiment** | | | | | | | | | |
| No. experiments | 10 | 10 | 10 | 10 | 10 | 10 | 10 | 10 | 3 |

## I.3 UCI Regression

We now provide the experimental details for the UCI regression experiments (as seen in Table 1 and Table 9). For each dataset, we ran a number of experiments to get a mean and standard deviation for performance metrics. In each experiment, we took a random $70\%/15\%/15\%$ split of the dataset for training, testing, and validation. The training hyper-parameters for the MAP, DE and NUQLS networks, size of the MLP used, and the number of experiments conducted for each dataset can be found in Table 12. Reported time for NUQLS, LLA and SWAG includes the training time for the original NN, with the time to run the method given in brackets.

- **NN:** For the PolyLR learning rate scheduler, a PyTorch polynomial learning rate scheduler was used, with parameters total_iters=$10\times$epochs, power$= 0.5$. The MLP used a $\tanh$ activation, so as to have smooth gradients. MLP weights were initialized as Xavier normal, and bias as standard normal. The dataset was normalized, so that the inputs and the outputs each had zero mean and unit standard deviation.

- **NUQLS:** The linear networks were trained using (S)GD with Nesterov momentum parameter $0.9$. For all datasets except for Song, the full training Jacobian could be stored in memory; this made training extremely fast. For the Song dataset, we trained all linear networks in parallel, by explicitly computing the gradient using JVPs and VJPs. The number of linear networks used was 10 across all datasets, and the $\gamma$ hyper-parameter was kept at $0.01$.

- **DE:** Each member of the ensemble output a separate heteroskedastic variance, and was trained to minimise the Guassian negative log likelihood. The ensemble weights were also initialized as Xavier normal, and bias as standard normal. The number of ensemble members was kept at 10.

- **LLA:** LLA requires two parameters for regression: a dataset noise parameter, and a prior variance on the parameters (Foong et al., 2019). To find the noise parameter, a grid search over 10 values on a log-scale between $1e-2$ and $1e2$ was used to find the noise that minimized the Gaussian likelihood of the validation set, with the LLA mean predictor as the Gaussian mean. The same grid search was used to find the prior variance, in order to minimize the expected calibration error (ECE) of LLA on the validation set. For the Protein dataset, a Kronecker-Factored Curvature (KFAC) covariance structure was used (Immer et al., 2021), and for the Song dataset a diagonal covariance structure was used. For all other datasets, LLA used the full covariance structure. The predictive distribution was computed with 1000 MC samples.

- **SWAG:** SGD was used, and the learning rate and number of epochs was kept the same as the NN. We used a grid-search for weight decay, over the values $[0, 0.005, 0.00005]$, to minimize the ECE on the validation set. The rank of the covariance matrix was 10 for all datasets, and the predictive distribution used 1000 MC samples.

Table 13: Training procedure for image classification results in Table 2, Figure 3, and Figure 6.

| MAP | LeNet5 MNIST | Lenet5 FMNIST | ResNet9 FMNIST | ResNet50 CIFAR10 | ResNet50 CIFAR100 | ResNet50 SVHN | ResNet50 ImageNet |
|---|---|---|---|---|---|---|---|
| Learning Rate | $5 \times 10^{-3}$ | $5 \times 10^{-3}$ | $10^{-3}$ | $10^{-2}$ | $10^{-1}$ | $10^{-2}$ | - |
| Epochs | 35 | 35 | 10 | 200 | 200 | 50 | - |
| Weight Decay | $10^{-4}$ | $10^{-4}$ | $10^{-4}$ | $10^{-4}$ | $5 \times 10^{-4}$ | $5 \times 10^{-4}$ | - |
| Batch Size | 152 | 152 | 100 | 128 | 128 | 128 | - |
| Optimizer | Adam | Adam | Adam | SGD | SGD | SGD | - |
| Scheduler | Cosine | Cosine | Cosine | Cosine | Cosine | Cosine | - |
| Accuracy | 99% | 90% | 92.5% | 92.5% | 75% | 87% | 76% |
| | | | | | | | |
| **NUQLS** | | | | | | | |
| Learning Rate | $10^{-2}$ | $10^{-2}$ | $10^{-1}$ | $10^{-1}$ | $10^{-2}$ | $10^{-2}$ | $10^{-2}$ |
| Epochs | 10 | 10 | 2 | 50 | 10 | 2 | 2 |
| Batch Size | 152 | 152 | 50 | 128 | 128 | 256 | 56 |
| $\gamma$ | 1 | 0.7 | 0.1 | 0.01 | 0.05 | 0.05 | 0.7 |

## I.4 Image Classification

We display the training procedure in Table 13 for both Figure 3 and Table 2. For MNIST and FashionMNIST, we took a $5:1$ training/validation split of the training data. For CIFAR-10, we simply used the test data as a validation set. For CIFAR-10, random horizontal crop and flip on the training images was used as regularization.

- **NN:** We chose the training procedures to provide the best MAP performance. All networks have weights initialized as Xavier uniform. For SGD, a momentum parameter of $0.9$ was used. For the Cosine Annealing learning rate scheduler, the maximum epochs was set to the training epochs.
- **NUQLS:** The number of samples was kept to 10 for all datasets.
- **DE:** Similarly, 10 ensemble members were used for all datasets.
- **MC-Dropout:** Dropout was applied to the network before the last fully connected layer. The dropout probability was set to $0.1$ (a larger probability of $0.25$ was also used, but it did not change the result). At test time, 100 MC-samples were taken.
- **SWAG:** The network was trained for a further $1\times$ training epochs. For CIFAR-100 and SVHN, learning rate was kept the same as the original NN; for other datasets, a larger learning rate of $1e2\times$NN learning rate was used. The covariance rank was set at 10. At test time, 100 MC-samples were taken.
- **LLA*:** We used a last-layer KFAC approximation to the covariance. The prior precision was found through a grid search over 20 values on a log scale from $1e-2$ to $1e2$, using the probit approximation to the predictive, and a validation set. This configuration for LLA is what is recommended in Daxberger et al. (2021). We used 1000 samples, to remedy the large amount of approximations used.
- **VaLLA**: We used the implimentation found in (Ortega et al., 2023). We kept nearly all hyper-parameters the same as in the MNIST and FashionMNIST experiments in (Ortega et al., 2023); however, we reduced the number of iterations to 5000, due to time constraints. We were unable to run this implimentation for ResNet9 or ResNet50, due to an out-of-memory error.

## I.5 Confidence Intervals

For the confidence intervals experiments in Table 4 and Table 3, we used 100 repeats for $d \leq 4$ and 50 repeats for $d \geq 8$. We used a multiplier of $1.5$ on all NUQLS variance values, as we note that the search strategy for $\gamma$ for regression only employs ECE; this does not reward confidence intervals that exceed the expected confidence, and hence computed confidence intervals may be conservative in width.

## I.6 Comparison Experiments

### I.6.1 BDE

For Figure 4 and Table 8, BDE was implimented following details given in (He et al., 2020). To make comparison with NUQLS fair, each ensemble member of BDE did *not* output a separate hetero-skedastic variance term. Instead, the variance over predictions was taken as the variance. Further, the large computational cost of BDE relative to NUQLS meant that it is unrealistic for comparison to allow BDE to tune hyper-parameters through a grid-search, as this would take the run time of BDE

(which is slightly greater than that of DE) severely out-of-proportion to NUQLS. For Figure 4, the noise-parameter was given as the noise in the data, 9. For Table 8, standard Gaussian noise was taken a priori. The ensemble members were trained with the same training scheme as the original NN in both Figure 4 and Table 8. For both experiments, 10 ensemble members were used. For classification, we again followed the implementation details given in He et al. (2020) to the best of our ability. We employed the same hyperparameters as DE/MAP, and used 10 ensemble members.

### I.6.2 SLU

For the results in Table 5, we copied the training details for (Miani et al., 2024a, Table 3.); we point the reader to (Miani et al., 2024a, Appendix D.1) for the exact details. For NUQLS, we used the following hyper-parameters for both MNIST and FashionMNIST: **Epochs** 10, **samples** 10, **learning rate** 0.01, **batch size** 152, $\gamma$ 1.

### I.6.3 Sampling-LLA, VaLLA and ELLA

For the results in Table 6, we used the training details for (Antorán et al., 2022, Figure 3. (left)); again, we point the reader to the details given in (Antorán et al., 2022, Appendix F.1). For NUQLS, we used the following hyper-parameters for MNIST: **Epochs** 10, **samples** 10, **learning rate** 0.01, **batch size** 152, $\gamma$ 0.25. For ResNet20 the following hyper-parameters were used: **Epochs** 1, **samples** 100, **learning rate** 0.0001, **batch size** 152, $\gamma$ 0.01. For ResNet32 the following hyper-parameters were used: **Epochs** 1, **samples** 150, **learning rate** 0.0001, **batch size** 152, $\gamma$ 0.01.

### I.6.4 BE

We employed the WideResNet architecture from Franchi et al. (2023). We used a depth of 10 and a widening factor of 1. The MAP network was trained with: **Epochs** 20, **learning rate** 0.005, **weight decay** 0.0001, **batch size** 152, the **Adam** optimizer and the **Cosine** learning rate scheduler. BE employed the same hyperparameters, yet with $1.5 \times$ **Epochs**, to aid with training. BE used 5 ensemble members, in order to match the computational budget of NUQLS. The NUQLS hyperparameters were: **Epochs** 10, **samples** 10, **learning rate** 0.01, **batch size** 152, $\gamma$ 1.

### I.6.5 SNGP

We employed the *Lightning UQ Box* (Lehmann et al., 2025) implementation of SNGP for our experiments. The same hyperparameters as the MAP network were used. Note that we also tested SNGP on ResNet50 SVHN, however SNGP was unable to train successfully, even after trying 3 learning rates and 2 optimizers. Hence, we did not include the results for ResNet50 SVHN.

### I.6.6 SGLD

We have implemented the SGLD method from Welling & Teh (2011) using the Lightning UQ Box package (Lehmann et al., 2025) as the basis of the code. We then amended this code to include the learning rate scheduler from Welling & Teh (2011). We followed Maddox et al. (2019), and initialized the SGLD trajectory from the weights of the trained network. We also copied the learning rate from Maddox et al. (2019), and used the same weight-decay as the original network for the SGLD prior. However, in contrast to Maddox et al. (2019), we took the noise-factor scaling to be $10^{-3}$ instead of $5 \times 10^{-4}$, as we found that this gave better performance. We sampled 100 epochs from the posterior, using a batch size of 100.

