# OpenReview forum: "Uncertainty Quantification with the Empirical Neural Tangent Kernel"
_NeurIPS.cc/2025/Conference — NeurIPS 2025 poster_

### Official Review · Reviewer_4xTc · 2025-06-30

**Clarity:** 3
**Significance:** 3
**Originality:** 3
**Rating:** 5
**Confidence:** 4

**Summary:**

This paper proposes a post-hoc ensembling method that approximately samples from the NTK GP posterior under certain conditions, mostly related to the convexity of the loss in the predictions of the network. The method linearizes the neural network, and reformulates the objective for the linearized network. This objective is optimized for multiple times from multiple randomized starting points that are obtained by distorting the original solution with Gaussian noise.

The method is motivated theoretically by Theorem 3.2 and Corollary 3.5, which roughly state that the proposed algorithm samples from the NTK GP posterior when the loss is convex.

The method is also tested on a toy regression task and several regression and classification data sets. The latter being interesting because the assumptions of Theorem 3.2 do not work, even though practical performance is still high. The authors test with established metrics they deem flawed as well as a newly proposed metric that creates violin plots of the variance of predictions.

**Questions:**

- Could the authors elaborate on the new UQ metric and its novelty? Is it the idea of taking the variance of the softmax outputs, or the act of visually inspecting the resulting violin plots?

- Can the authors elaborate on the computations described in appendix D, perhaps in mathematical forms?

**Ethical Concerns:**

["NO or VERY MINOR ethics concerns only"]

**Final Justification:**

The authors have fully addressed my questions in their rebuttal to my review and in the rebuttal to j2t5. They have also added multiple baselines that convince me that the experimental results are sound and that the conclusions based on them are reliable. I feel that my original rating of 5 accurately reflects the quality of the paper.

**Limitations:**

Limitations are addressed adequately. The authors acknowledge that their theoretical motivation relies on assumptions that are often violated, which is common in NTK theory.

**Paper Formatting Concerns:**

No concerns.

**Quality:**

3

**Strengths And Weaknesses:**

This is a good paper. It is well written and easy to read. The introduction and background gives a good overview of current literature, although I was missing perhaps gradient based MCMC methods which also have been applied to neural networks, see e.g. [1-3].

The theoretical justification is laid out well and not too difficult to follow. Such theoretical justifications are also especially important for uncertainty quantification which is inherently difficult to benchmark in cases where we do not have access to the true posterior. I resonate strongly with the messages in Appendix C about flaws in common benchmarking metrics, although I can't speak to the novelty of the proposed UQ metric.

The experiments are well designed and the algorithm shows good performance on established (but possibly flawed) metrics as well as the proposed UQ metric. I understand the reasoning to exclude BDE, but do not understand how the same reasoning would not also apply to DE. I think including BDE in the image classification experiments could benefit the clarity and transparency of the paper, and it would also be interesting to see how BDE compares to DE on the newly proposed metric. Finally, I would also have liked to see an SGMCMC method in the baselines.

Overall, this is a very interesting work, and uncertainty quantification for neural networks is an important research topic. I believe the field would benefit from this paper at a high profile venue like NeurIPS.

[1] Wenzel, Florian, et al. "How good is the bayes posterior in deep neural networks really?." _arXiv preprint arXiv:2002.02405_ (2020).

[2] Izmailov, Pavel, et al. "What are Bayesian neural network posteriors really like?." _International conference on machine learning_. PMLR, 2021

[3] Garriga-Alonso, Adrià, and Vincent Fortuin. "Exact langevin dynamics with stochastic gradients." _arXiv preprint arXiv:2102.01691_ (2021).

---

> ### Author Rebuttal · Authors · 2025-07-31
>
> > I was missing perhaps gradient based MCMC methods which also have been applied to neural networks, see e.g. [1-3]...I would also have liked to see an SGMCMC method in the baselines.
>
> We thank the reviewer for bringing attention to the SGMCMC methods. We originally did not compare to any of these methods due to the often large cost associated with computing them in high-dimensions, and due to some poor performance in previous UQ benchmarks (albiet computed using flawed metrics). However, we appreciate the importance of SGMCMC to the Bayesian framework, in that they are able to compute approximate samples from the posterior. We are actively working toward addressing your comment and will endeavour to include an SGMCMC method in our comparisons before the end of the rebuttal period.
>
>
> >  I understand the reasoning to exclude BDE, but do not understand how the same reasoning would not also apply to DE. I think including BDE in the image classification experiments could benefit the clarity and transparency of the paper, and it would also be interesting to see how BDE compares to DE on the newly proposed metric.
>
> This is certainly an interesting point, and we appreciate the opportunity to clarify. We find that the prevailing opinion in current Bayesian UQ works is that DE is still the SOTA, even with the addition of BDE. Further, we find it rare that works will compare with BDE, rather than simply comparing with DE. Hence, we currently rank DE as the highest performing method. A primary goal of our work was to achieve performance similar to DE, whilst reducing the computational cost. We then sought to compare with other methods that seek to achieve a similar goal. Hence, this is why we excluded BDE from the image classification results. However, we agree that a comparison of BDE to both DE and NUQLS on this new metric is enlightening, as BDE may in fact show better performance than DE. Further, the similarity / difference between BDE and NUQLS in this setting will be interesting to observe. Thus, we will aim to evaluate BDE on an image classification result and hope to provide the results before the end of the rebuttal author-reviewer period.
>
> > Could the authors elaborate on the new UQ metric and its novelty? Is it the idea of taking the variance of the softmax outputs, or the act of visually inspecting the resulting violin plots?
>
> This is a great question, and we are happy to provide further details on this point that we may have skipped over in the main text. We firstly find that no previous works have employed the variance of the softmax outputs as the quantity used to compare UQ performance. For example, most works simply use the softmax outputs themselves to form evaluation metrics, either by taking the maximum class prediction as the confidence, or the entropy over the classes. The SLU work [4] used the variance of the logits to compute the AUCROC between ID and OOD points (see Appendix G.1 in our work for a small comparison to SLU on this metric). However, we are unaware of other works that have employed the variance of the softmax outputs. Secondly, previous works, such as SWAG (Appendix E.2), have plotted the entropic histograms of the softmax predictions for both ID and OOD points. To our knowledge though, we are the first work which employs violin plots, that compares the median and skew values, and that compares to a poorly performing baseline. Hence, we see that the metric is novel in both the quantity it uses for evaluation, and the manner in which it compares performance between methods. Note that we also provide further details on the VMSP metric in response to a question from reviewer j2t5.
>
> > Can the authors elaborate on the computations described in appendix D, perhaps in mathematical forms?
>
> We are happy to provide further mathematical details on the computations in Appendix D. We compare the computational complexity for an epoch of training for the neural network $f_\theta(x)$, for batch $x \in \mathbb{R}^{d \times n}$, and an epoch of training for a single linear network $\hat{f}\_\theta(x)$. We take approximations on the computational complexity of both forward-mode AD and backward-mode AD from the following text [5]. Specifically, we take $[fp]$ as the computational complexity for evaluating $f_\theta(x)$.
>
> We note from [5] that both a JVP and a VJP cost roughly $2$ $\times$ a forward pass in computational complexity and memory. Further, both a JVP and a JVP return a function evaluation. Now, a standard epoch of training for the neural network involves a forward-pass to compute the error, and then a backward-pass (VJP), hence the complexity is approx. $3[fp]$. The linearised network involves a JVP (which includes a function evaluation) to form the linear network, and a VJP to compute the gradient. Hence, the complexity for an epoch of training for the linearized network is approx. $4[fp]$. So we observe that each epoch for a linearized network is only $4/3~\times$ as expensive as for a neural network.  In regards memory, we see that the memory requirement for both the linearized network and the neural network are similar. However, we generally train all linearized networks in parallel. For computational complexity, this will incur some additional cost, thought it will not be linear in number of networks, that is dependent upon the specific software and hardware. However, memory cost will scale linearly by number of networks, i.e. $2S \times M([fp])$, where $M([fp])$ is the memory cost for a forward-pass, and $S$ is the number of linear networks. As an example, we employ a batch size of $56$ for ImageNet on ResNet50 with $10$ ensemble members when using an 80GB H100 GPU, due to the large parameter count and large number of classes.
>
> These improvements have been made to the working copy of our paper (pending additional experiments). Please let us know if you have any other questions or points that need clarification.
>
> ---
> References:
>
> [1] Wenzel, Florian, et al. "How good is the bayes posterior in deep neural networks really?." arXiv preprint arXiv:2002.02405 (2020).
>
> [2] Izmailov, Pavel, et al. "What are Bayesian neural network posteriors really like?." International conference on machine learning. PMLR, 2021
>
> [3] Garriga-Alonso, Adrià, and Vincent Fortuin. "Exact langevin dynamics with stochastic gradients." arXiv preprint arXiv:2102.01691 (2021).
>
> [4] Sketched Lanczos uncertainty score: a low-memory summary of the Fisher information, Miani et. al. 2024
>
> [5] The Elements of Differentiable Programming, Blondel et. al. 2024, Chapter 8

---

> > ### Author Response · Authors · 2025-08-06
> >
> > We hope that we have sufficiently addressed the concerns of the reviewer with the inclusion of the Bayesian Deep Ensemble comparison. Further, we are still working to provide a comparison of our method NUQLS with an SGMCMC method (specifically SGLD) before the end of the review period. We would be glad to engage in further discussion if you have any further questions or points that require clarification regarding our paper.

---

> > > ### Comment · Reviewer_4xTc · 2025-08-06
> > >
> > > I thank the authors for their rebuttal. I appreciate the discussion regarding the metric, including their conversation with reviewer j2t5. I also acknowledge their addition of BDE to the baselines in their general comment.
> > >
> > > Their explanation of computational complexity was clarifying.
> > >
> > > I commend the authors for their efforts of adding an SGMCMC method to the baselines, and agree that SGLD is a valid choice. I am looking forward to these results and believe that they would strengthen the paper.
> > >
> > > I have no further questions at this point.

---

> > > > ### Author Response · Authors · 2025-08-08
> > > >
> > > > We are happy to be able to provide a comparison of NUQLS with SGLD, using a ResNet9 network trained on FashionMNIST. We have implemented the SGLD method from [1] using the Lightning UQ Box package [2] as the basis of the code. We then amended this code to include the learning rate scheduler from [1]. We followed the SWAG paper [3], and initialized the SGLD trajectory from the weights of the trained network. We also copied the learning rate from [3], and used the same weight-decay as the original network for the SGLD prior. However, in contrast to [3], we took the noise-factor scaling to be $10^{-3}$ instead of $5\times10^{-4}$, as we found that this gave better performance. We sampled $100$ epochs from the posterior, using a batch size of $100$. We present the values in the following tables, in comparison to the other methods we have implemented. The median values are:
> > > >
> > > > |                | NUQLS                  | DE                    | BDE                    | SGLD                  | SWAG                  | MC                    | LLA                    | BASE   |
> > > > |----------------|------------------------|-----------------------|------------------------|-----------------------|-----------------------|-----------------------|------------------------|--------|
> > > > | R9FM: IDC (<)  | $4.76 \times 10^{-15}$ | $1.31 \times 10^{-5}$ | $3.35 \times 10^{-20}$ | $6.79 \times 10^{-8}$ | $4.64 \times 10^{-7}$ | $5.05 \times 10^{-7}$ | $9.02 \times 10^{-10}$ | $0.0206$ |
> > > > | R9FM: IDIC (>) | $0.182$                | $0.0397$              | $0.00915$              | $0.023$              | $0.0478 $             | $0.00413$             | $0.00372$              | $0.0199$ |
> > > > | R9FM: OOD (>)  | $0.217$                | $0.109$               | $0.0504$               | $0.0352$              | $0.0748 $             | $0.00447$             | $0.0086$               | $0.0199$ |
> > > >
> > > > The skew values are:
> > > >
> > > > |                | NUQLS | DE     | BDE     | SGLD  | SWAG   | MC      | LLA     | BASE   |
> > > > |----------------|-------|--------|---------|-------|--------|---------|---------|--------|
> > > > | R9FM: IDC (>)  | $2.4$ | $3.51$ | $6.46$  | $5.49$ | $3.72$| $4.92$   | $6.46$   | $0.93$ |
> > > > | R9FM: IDIC (<) | $-0.615$ | $0.928$ | $1.43$ | $1.23$ | $0.378$ | $1.6$ | $1.18$ | $1.15$ |
> > > > | R9FM: OOD (<)  | $-1.7$ | $0.321$  | $0.515$  | $1.1$ | $0.0756$ | $1.69$ | $0.926$  | $1.05$ |
> > > >
> > > > We see that SGLD performs very similarly to SWAG in this metric, which is to be expected, as they are both methods that are based on sampling from the SGD trajectory. However, we see that NUQLS shows superior performance to SGLD.
> > > >
> > > > We appreciate the suggestion to extend the comparison of NUQLS to both SGLD and BDE (as well as the suggestion of the other reviewers to compare to methods such as BatchEnsemble, SNGP and PNC), as we feel that it has strengthened the value of the experimental results.
> > > >
> > > > ---
> > > > References:
> > > >
> > > > [1] Bayesian Learning via Stochastic Gradient Langevin Dynamics, Welling et. al. 2011
> > > >
> > > > [2] Lightning UQ Box: A Comprehensive Framework for Uncertainty Quantification in Deep Learning, Lehmann et. al. 2024
> > > >
> > > > [3] A Simple Baseline for Bayesian Uncertainty in Deep Learning, Maddox et. al. 2019

---

### Official Review · Reviewer_CkYf · 2025-07-01

**Clarity:** 3
**Significance:** 2
**Originality:** 3
**Rating:** 4
**Confidence:** 4

**Summary:**

This paper investigates uncertainty quantification in neural networks. Specifically, it constructs an ensemble of linear models centered around the parameters of a trained deep neural network (DNN). To generate diverse initializations, isotropic Gaussian noise is added to the trained parameters. The paper presents a theoretical analysis showing that, under a regression loss, the proposed method samples models from an approximate posterior of the neural network, which corresponds to a Gaussian process defined by the neural tangent kernel. Empirically, the method demonstrates improved calibration of uncertainty estimates compared to baseline approaches on both regression and classification tasks.

**Questions:**

Since all linearized models are initialized around the same local optimum, is there a risk of mode collapse—i.e., all models converging to similar behavior? A discussion of this potential limitation and how it is mitigated would be helpful.

**Ethical Concerns:**

["NO or VERY MINOR ethics concerns only"]

**Final Justification:**

The author has provided additional experimental results to address my initial concerns during the rebuttal. So I think a weak accept is appropriate for this paper.

**Limitations:**

yes

**Paper Formatting Concerns:**

no issue

**Quality:**

3

**Strengths And Weaknesses:**

**Strengths:**

- Uncertainty quantification is a critical and largely unsolved challenge in deep learning. While Bayesian neural networks are principled, they often underperform in practice, and deep ensembles, though effective, are computationally expensive. The proposed method offers a compelling compromise between these two extremes.

- The paper is clearly written and well-structured.

- Experimental results are promising and demonstrate improved calibration over baselines on multiple tasks.

**Weaknesses:**

- The comparison to prior work is limited. Several efficient ensemble methods, such as BatchEnsemble [1], Snapshot Ensembles [2] were proposed. SNGP [3] are highly relevant but not included in the evaluation. Additionally, LLA is not a particularly strong Bayesian baseline.

- Although the paper evaluates both regression and classification tasks, it would strengthen the contribution to demonstrate that improved uncertainty estimates lead to better downstream performance (e.g., in active learning or decision-making under uncertainty).

- Prior work has studied the connection between uncertainty quantification and the neural tangent kernel (NTK). It would be helpful for the authors to clarify how their method differs from or extends these works, such as [4].

**References:**

[1] *BatchEnsemble: An Alternative Approach to Efficient Ensemble and Lifelong Learning*, ICLR 2020
[2] *Training Independent Subnetworks for Robust Prediction*, ICLR 2021
[3] *Simple and Principled Uncertainty Estimation with Deterministic Deep Learning via Distance Awareness*, NeurIPS 2022
[4] *Efficient Uncertainty Quantification and Reduction for Over-Parameterized Neural Networks*, NeurIPS 2023

---

> ### Author Rebuttal · Authors · 2025-07-31
>
> We thank the reviewer for their careful reading of our work, and constructive criticism.
>
> > The comparison to prior work is limited. Several efficient ensemble methods, such as BatchEnsemble [1], Snapshot Ensembles [2] were proposed. SNGP [3] are highly relevant but not included in the evaluation. Additionally, LLA is not a particularly strong Bayesian baseline.
>
> We thank the reviewer for bringing attention to the importance of these methods. Unfortunately due to the limited time of the rebuttal period, we were only able to impliment one of the referenced methods, of which we chose SNGP, as there exists a Lightning Pytorch implementation of the method [5]. We tested SNGP using VMSP on both ResNet9 / FashionMNIST and ResNet50 / SVHN. The median results are presented in the following table. We write $(>)$ to denote values which we want to be large, and $(<)$ for values we want to be small (synonymous with the up and down arrows in the paper, respectively).
>
> |               | NUQLS                  | SNGP                  | BASE    |
> |---------------|------------------------|-----------------------|---------|
> | R9FM: IDC (<)      | $4.76 \times 10^{-15}$ | $9.19 \times 10^{-7}$ | $0.020$ |
> | R9FM: IDIC (>)    | $0.182$                | $3.22 \times 10^{-4}$ | $0.020$ |
> | R9FM: OOD (>)     | $0.217$                | $3.42 \times 10^{-4}$ | $0.020$ |
> | R50SVHN: IDC (<)   | $8.61 \times 10^{-15}$ | $3.64 \times 10^{-6}$ | $0.020$ |
> | R50SVHN: IDIC (>) | $0.217$                | $3.64 \times 10^{-6}$ | $0.020$ |
> | R50SVHN: OOD (>)  | $0.233$                | $3.66 \times 10^{-6}$  | $0.020$ |
>
> and the skew results:
>
> |               | NUQLS    | SNGP   | BASE    |
> |---------------|----------|--------|---------|
> | R9FM: IDC (>)      | $2.40$   | $29.8$ | $1.08$  |
> | R9FM: IDIC (<)    | $-0.615$ | $24.4$ | $1.09$  |
> | R9FM: OOD (<)     | $-1.70$  | $1.21$ | $1.06$  |
> | R50SVHN: IDC (>)  | $2.2$    | $0.309$ | $0.020$ |
> | R50SVHN: IDIC (<) | $-1.31$  | $0.284$ | $0.020$ |
> | R50SVHN: OOD (<)  | $-1.78$  | $0.304$ | $0.020$ |
>
> We see that on ResNet9/FashionMNIST, NUQLS far out-performs SNGP. For ResNet50 / SVHN, $3$ learning rates and $2$ optimizers were tested; unfortunately SNGP did not properly train for the tested combinations, and hence the best results are very poor.
>
> Note that we focused our comparison on LLA variants as they have become popular recently, with the release of the Laplace package [5], and through scalable variants such as Sampling-LLA and VaLLA.
>
> We believe that the efficient ensemble methods are also important methods to consider, and hence we are working toward including results comparing these methods to NUQLS before the end of the review period.
>
> > Although the paper evaluates both regression and classification tasks, it would strengthen the contribution to demonstrate that improved uncertainty estimates lead to better downstream performance (e.g., in active learning or decision-making under uncertainty).
>
> We appreciate the suggestion of the reviewer to display the performance of NUQLS on a downstream task, as we believe the results would be interesting and would strengthen the value of the paper. Unfortunately we are unable to run a well-thought out, large scale experiment in the rebuttal period due to time constraints. However, we will include such an evaluation in future work.
>
> > Prior work has studied the connection between uncertainty quantification and the neural tangent kernel (NTK). It would be helpful for the authors to clarify how their method differs from or extends these works, such as [4].
>
> The referenced work PNC is very interesting, and the connection of NUQLS to PNC will be included in the paper. We refer the reviewer to our response to Reviewer Z6xt, where we discuss the inherent differences between NUQLS and PNC, as well as empirically compare them on an experiment from [4]. To summarise, PNC is developed under the asymptotic NTK theory and performs well for very wide networks near initialization (i.e., poorly trained networks), where NUQLS struggles. In contrast, NUQLS does not rely on the asymptotic NTK regime and instead leverages the trained, data-dependent NTK to construct meaningful linear ensembles. For finite-width networks that are well trained—which we believe to be the practically relevant setting—NUQLS is more appropriate and clearly outperforms PNC.
>
>
> > Since all linearized models are initialized around the same local optimum, is there a risk of mode collapse—i.e., all models converging to similar behavior? A discussion of this potential limitation and how it is mitigated would be helpful.
>
> This is an excellent question, and we're glad to offer a concrete answer: **mode collapse cannot occur—almost surely**.
>
> To see this, note that the solution to equation (3) in our main paper for each linearized model consists of a unique row-space component (under our assumptions), plus a projection of the initialization $z_i$ onto the null space of the Jacobian. For mode collapse to occur, the projections for two i.i.d. initializations $z_1, z_2 \sim \mathcal{N}(0, \gamma^2 I)$ would need to coincide—that is, we would require $(I - J^{\dagger}J)(z_1 - z_2) = 0$, even though $z_1 - z_2 \neq 0$ almost surely. This implies $z_1 - z_2 \in \text{Range}(J^T)$. However, $\text{Range}(J^T)$ is a low-dimensional subspace of $\mathbb{R}^p$, while $z_1$ and $z_2$ are drawn iid from full-rank (i.e., non-degenerate) $p$-dimensional Gaussian distribution. The probability that $z_1 - z_2$ lies in a fixed lower-dimensional subspace is zero. Therefore, mode collapse occurs with probability zero, and we can safely rule it out as a concern.
>
>
> We hope that this has alleviated your concerns with our work. Please let us know if there are any further questions of points that require clarification, as we would like to engage further if it will improve your assessment of our paper.
>
> ---
> References:
>
> [1] BatchEnsemble: An Alternative Approach to Efficient Ensemble and Lifelong Learning, ICLR 2020
>
> [2] Training Independent Subnetworks for Robust Prediction, ICLR 2021
>
> [3] Simple and Principled Uncertainty Estimation with Deterministic Deep Learning via Distance Awareness, NeurIPS 2022
>
> [4] Efficient Uncertainty Quantification and Reduction for Over-Parameterized Neural Networks, NeurIPS 2023
>
> [5] Lightning UQ Box: A Comprehensive Framework for Uncertainty Quantification in Deep Learning, Lehmann et. al. 2024
>
> [6] Laplace Redux – Effortless Bayesian Deep Learning, Daxberger 2021

---

> > ### Comment · Reviewer_CkYf · 2025-08-04
> >
> > Thanks the author for the detailed responses. I do not have any further questions. I hope the author could include the results they promised in the final version.

---

### Official Review · Reviewer_Z6xt · 2025-07-03

**Clarity:** 2
**Significance:** 2
**Originality:** 2
**Rating:** 3
**Confidence:** 3

**Summary:**

The paper proposes a post-hoc, sampling-based uncertainty quantificaiton method for overparameterized networks by approximating the predictive distribution of deep models using Gaussian Processes (GPs) by neural tanget kernels. The proposed method, NUQLS first update the learned parameters $\hat{\theta}$ with $S$ steps for the posterior (9). The method has lower computational complexity than full Bayesian neural networks (BNNs) and avoids costly ensemble-based inference.The authors demonstrate the performance of their method on regression and classification benchmarks, comparing against deep ensemble, SWAG, MC, and LLA.

**Questions:**

- Robustness to Representation Quality:
How sensitive is your method to the quality of the learned representations? Would randomly initialized or undertrained networks still yield calibrated uncertainty?

-Non-Gaussian Outputs:
How do you handle classification tasks where predictive distributions are inherently non-Gaussian (e.g., multimodal posteriors)? Does your method underperform in such settings?

**Ethical Concerns:**

["NO or VERY MINOR ethics concerns only"]

**Final Justification:**

While the empirical performance of the method is strong, the theoretical justification is somewhat unclear. In particular, Lemma 3.1 is derived using convexity, but its proof appears to be questionable. The use of convex loss to argue for a unique solution in a local region also seems unnecessary. Given these issues, I decided to keep the score.

**Limitations:**

The method’s performance is evaluated primarily on small to medium-scale datasets, limiting the generalizability of its claims to larger or more diverse real-world scenarios.

**Quality:**

2

**Strengths And Weaknesses:**

Strengths
1.	The idea of using random features to approximate a GP on top of neural network embeddings is simple.
2.	The paper evaluates expected calibration error (ECE), negative log-likelihood (NLL), and predictive intervals on standard datasets such as UCI benchmarks and CIFAR10.
Weaknesses
1.	Unlike the NeurIPS 2023 PNC paper, the submission lacks rigorous statistical theory (e.g., asymptotic coverage guarantees, bias-variance decomposition) to back the empirical results. Additionally, compared to the SWAG paper, insufficient empirical evidence is provided to demonstrate how Equation (9) (the main result) relates to uncertainty quantification.
2.	The method relies on approximate Gaussianity of the hidden-layer representations' output distribution and the strong convexity of the loss function. These assumptions may break down for complex real-world classification tasks (especially with multimodal outputs), yet this limitation is not critically analyzed.
3.	There is little discussion on when or why the method might fail (e.g., poorly trained base networks, overfitting, high-dimensional noise in embeddings).

---

> ### Author Rebuttal · Authors · 2025-07-31
>
> >  Unlike the NeurIPS 2023 PNC paper, the submission lacks rigorous statistical theory (e.g., asymptotic coverage guarantees, bias-variance decomposition) to back the empirical results.
>
> We appreciate the opportunity to highlight the differences between the PNC paper and ours. Firstly, PNC is formed from a frequentist framework, and hence, similar to conformal predictions, they are able to get statistical guarantees on confidence interval coverage. Further, PNC relies on the NTK theory, where for infinitely wide networks, the network behaves according to kernel regression. From this, the authors are able to estimate possible function values. Instead, NUQLS relies on the network being trained to a local minima, and then employs the feature-learning of finite-width networks to create meaningful ensembles.
>
> To test the difference between the two methods, we tested NUQLS on the _confidence intervals_ problem from [3]. In this experiment, an MLP with a single hidden-layer is trained on $n$ datapoints from $U([0,0.2]^d)$, where the target is $y = \sum_{i=1}^d \sin(x_i) + \epsilon$, and $\epsilon$ is a small noise term. The method is then asked to form a confidence interval around the prediction for $x = (0.1, \dots, 0.1)$. This setup is repeated several times, and the coverage and width of the confidence intervals are recorded, as well as the mean prediction. For more details, we refer the reviewer to [3]. When the width of the network is $32 \times n$, and the network is only partially trained with $80$ epochs and a learning rate of $0.01$, NUQLS performs poorly compared to PNC, as can be seen in the following table. Note that bolded numbers indicate intervals that have reached or exceeded the expected coverage. Further, smaller width intervals are preferred.
>
>
> |                 |      PNC     |              |        |     NUQLS    |              |        |
> |:---------------:|:------------:|:------------:|:------:|:------------:|:------------:|:------:|
> |                 | 95\%CI (CR/IW) | 90\%CI (CR/IW) |   MP   | 95\%CI (CR/IW) | 90\%CI (CR/IW) |   MP   |
> | (d = 2) n = 128 |  **0.98**/0.0437 |  **0.95**/0.0323 | 0.1998 |  0.93/0.0357 |  **0.92**/0.0299 | 0.2047 |
> | (d = 4) n = 256 |  **0.98**/0.0411 |  **0.95**/0.0304 | 0.3991 |  0.92/0.0596 |  0.86/0.0500 | 0.4084 |
>
> We note that scaling the width of a network by $32$ times the number of training points is rare in practice, and that such a wide-network is difficult to train. If we instead form an MLP with width equal to the number of training points, and increase epochs to $1000$ and learning rate to $0.5$, so that the network is properly trained, NUQLS far outperforms PNC, as can be seen in the following table. We note that for these experiments we tuned the $\gamma$ hyper-parameter for NUQLS on a small validation set.
>
> |                  |      PNC     |              |        |     NUQLS    |              |        |
> |:----------------:|:------------:|:------------:|:------:|:------------:|:------------:|:------:|
> |                  | 95\%CI (CR/IW) | 90\%CI (CR/IW) |   MP   | 95\%CI (CR/IW) | 90\%CI (CR/IW) |   MP   |
> |  (d = 2) n = 128 |   0.8/0.136  |  0.72/0.0105 | 0.2022 |  **0.99**/0.0135 |  **0.96**/0.0114 | 0.2012 |
> |  (d = 4) n = 256 |  **0.96**/0.0437 |  **0.9**/0.0336  | 0.4045 |  **0.97**/0.0313  |  **0.92**/0.0263 | 0.4030 |
> |  (d = 8) n = 512 |  0.88/0.0740 |  0.88/0.0568 | 0.8078 |  **1.00**/0.0667 |  **0.98**/0.0559 | 0.8050 |
> | (d = 16) n = 128 |  0.8/0.1443  |  0.8/0.1108  | 1.6265 |  **1.00**/0.1350 |  **0.98**/0.1133 | 1.6121 |
>
> We observe that NUQLS and PNC are in fact complementary methods. For infinite-width networks near initialization, PNC performs well while NUQLS struggles. Conversely, for finite-width networks trained to a minimum, NUQLS excels while PNC performs poorly. We believe the latter regime is more representative of practical scenarios, where NUQLS offers significantly better performance.
>
> Finally, we note that rigorous statistical guarantees that exist for PNC do not currently exist for Bayesian methods, a point that is raised in the PNC paper [3].
>
> > Compared to the SWAG paper, insufficient empirical evidence is provided to demonstrate how Equation (9) (the main result) relates to uncertainty quantification
>
> In regards to the experiments in the submitted paper, we find that under a variety of regression and classification tasks, our method does in fact accurately quantify the uncertainty in the predictions of a neural network. Further, we have now also provided an extra regression experiment, i.e. the _confidence intervals_ experiment from [3], and a large-scale image classification experiment, i.e. the **ResNet50/ImageNet** result that has been provided in the response to Reviewer j2t5. From this, we hope that the reviewer finds adequate empirical evidence that the variance term derived in (9) does in fact accurately quantify the uncertainty in a neural networks predictions.
>
> > The method relies on approximate Gaussianity of the hidden-layer representations' output distribution and the strong convexity of the loss function. These assumptions may break down for complex real-world classification tasks (especially with multimodal outputs), yet this limitation is not critically analyzed...How do you handle classification tasks where predictive distributions are inherently non-Gaussian (e.g., multimodal posteriors)? Does your method underperform in such settings?
>
> We would like to respectfully clarify a potential misunderstanding: we do not assume Gaussianity for the original neural network at any point. Rather, if the network is approximated by its linearization—a reasonable assumption near a local minimum—we show that the predictive distribution of the linearized model follows a Gaussian Process. Consequently, the predictive distribution of the original network is approximately Gaussian in this regime.
>
> Even in real-world classification tasks such as ImageNet with ResNet-50, where the posterior is likely multimodal, we observe excellent uncertainty quantification. While we have no guarantee that the predictive distribution is exactly Gaussian in these settings, we find that the linearized networks form meaningful ensembles that effectively explore the local function space and capture predictive uncertainty.
>
>
> > There is little discussion on when or why the method might fail (e.g., poorly trained base networks, overfitting, high-dimensional noise in embeddings)...How sensitive is your method to the quality of the learned representations? Would randomly initialized or undertrained networks still yield calibrated uncertainty?
>
> This is an insightful question, and has prompted an important addition to our paper. As has been previously noted, we can see from the attached _confidence interval_  results that untrained or undertrained networks do not necessarily yield well-calibrated uncertainty estimates. Instead, if the network is very wide and poorly trained, a method like PNC would be more appropriate. However, we again note that this would be uncommon in practice, as most networks will be well-trained. In the case of well-trained networks, we find that NUQLS excels.
>
> > The method’s performance is evaluated primarily on small to medium-scale datasets, limiting the generalizability of its claims to larger or more diverse real-world scenarios.
>
> We kindly refer the reviewer to our response to Reviewer j2t5, where we show results for ResNet50 trained on **ImageNet**. This setting is generally considered large-scale in the Bayesian UQ commmunity.
>
> Please let us know if you have any questions or there is something else that needs clarification, as we are eager to increase your assessment of our work.
>
> ---
> References:
>
> [1] Deep learning versus kernel learning: an empirical
> study of loss landscape geometry and the time
> evolution of the Neural Tangent Kernel, NeurIPS2020
>
> [2] Harnessing the Power of Infinitely Wide Deep Nets on Small-data
> Tasks, Arora et. al. 2019.
>
> [3] Efficient Uncertainty Quantification and Reduction for
> Over-Parameterized Neural Networks, Huang et. al. 2023

---

> > ### Author Response · Authors · 2025-08-06
> >
> > In view of the recent instructions from the Program Committee, we would like to invite further discussion on any remaining points that may need clarification. We believe that we have provided satisfactory responses to the concerns of the reviewer during the rebuttal. Specifically, we have provided further empirical results that demonstrated the scalability of our method, and its favourable comparison to the PNC method, as well as given explanations as to the limitations of our method and the applicability to classification tasks.

---

> > > ### Comment · Reviewer_Z6xt · 2025-08-06
> > >
> > > I appreciate the reviewer’s detailed reading of our response.
> > >
> > > First, I acknowledge the reviewer’s valuable point regarding the differences between PNC and NUQLS, particularly on the regime where PNC, derived from NTK-based analysis, performs well for extremely wide and near-initialized networks, while NUQLS benefits from feature learning in practically trained finite-width networks.
> > > I agree that these methods are complementary rather than directly competing.
> > >
> > > Regarding the Gaussianity and non-convexity concern, I agree the explanation on Taylor's approximation in the neighborhood of the local minima.
> > > However, the equation (5) is considered as a strongly convex function, which seems to be a strong assumption for the parameter $\theta$, and the derivation in lines 247-256 appears to be not enough to support the author's claim.
> > >
> > > In addition, I thank the reviewer for providing more results on Imagenet.
> > > However, I still have a concern on the above theoretical issues.

---

> > > > ### Author Response · Authors · 2025-08-07
> > > >
> > > > We appreciate the opportunity for further discussion. Regarding your concern about the strong convexity assumption on the loss function in Eq. (5), we would like to clarify a few potential misunderstandings.
> > > >
> > > > It appears there may have been a misinterpretation of the assumptions underlying Eq. (5). Specifically, the equation does not require strong convexity with respect to the parameter $\mathbf{\theta}$. Rather, it only assumes that the loss is strongly convex in $\mathbf{\tilde{f}}$, or strictly convex when a solution exists. This condition is satisfied in standard settings, for example, in regression with the squared-error loss, or in classification using the Brier loss. We note that models trained with the Brier loss (a strongly convex loss commonly used in classification) have been shown to perform comparably to those trained with cross-entropy [1], and is often used as a proxy for theoretical purposes. We will clarify this in Remark 3.6 in the final version of the document.
> > > >
> > > > Importantly, these assumptions on the loss function are only needed for deriving the explicit GP predictive form in Eq. (9). For widely used loss functions that do not meet these assumptions (e.g., the cross-entropy loss, as discussed in Remark 3.6), the derivations in lines 247-256 do not apply. Nevertheless, we still form an ensemble of locally linearised models based on the empirical NTK, each trained to minimize the training loss. This provides a meaningful ensemble of high-performing linear surrogates, from which we can compute the empirical variance of a prediction. Our extensive experiments show that even in image classification tasks using the strictly convex cross-entropy loss, NUQLS performs on par with, or even outperforms, its competitors.
> > > >
> > > > ---
> > > > References:
> > > > [1] Evaluation of Neural Architectures Trained with Square Loss vs Cross-Entropy in Classification Tasks, Hui et. al. 2020

---

### Official Review · Reviewer_j2t5 · 2025-07-06

**Clarity:** 2
**Significance:** 2
**Originality:** 2
**Rating:** 4
**Confidence:** 3

**Summary:**

The paper proposes NUQLS (Neural Uncertainty Quantification by Linearised Sampling), a post-hoc method that draws an ensemble of linearised networks around trained parameters and then runs (stochastic) gradient descent to (approximately) solve a convex surrogate problem. Under stated conditions, the resulting predictors are shown to be samples from the predictive posterior of a Gaussian process with the empirical neural-tangent-kernel (NTK). Experiments on toy regression, nine UCI regression sets, and image-classification benchmarks (FashionMNIST, CIFAR10/100) show that NUQLS matches or outperforms Deep Ensembles, Laplace/LLA variants, SWAG, and MC-Dropout while being up to 10 faster in wall-clock time on the larger tasks than Deep Ensembles. The paper also introduces a variance-of-mean-softmax-probabilities (VMSP) metric for classification uncertainty.

**Questions:**

1. Can you demonstrate scalability to larger, more modern architectures? And additional datasets with additional epistemic uncertainty metrics?
2. How does the method perform when the linearization approximation quality degrades?
3. Can you compare to the most important related work in the main paper?

**Ethical Concerns:**

["NO or VERY MINOR ethics concerns only"]

**Final Justification:**

The authors have provided a good rebuttal. I'm still split on the novelty, but I'm also not sure if I know the literature well enough. The point about sampling from the null space is not novel by itself: Miani, Marco, Hrittik Roy, and Søren Hauberg. "Bayes without Underfitting: Fully Correlated Deep Learning Posteriors via Alternating Projections." AISTATS, 2025, ArXiv Oct 2024, which also projects onto the kernel, and which I'll add to my response. Unlike that paper, this paper remains really hard to understand, though.

**Limitations:**

The authors adequately discuss computational limitations and the restricted theoretical scope. However, they could better address the theory-practice disconnect and provide more analysis of when the linearization approximation might fail.

**Quality:**

3

**Strengths And Weaknesses:**

### **Strengths**

**Computational Efficiency**: The method demonstrates significant computational speedups over deep ensembles (often order-of-magnitude improvements) while maintaining competitive performance on the tested datasets.

**Clear Algorithmic Contribution**: Algorithm 1 is simple, interpretable, and appears to be readily implementable with minimal modifications to existing training pipelines.

**Empirical Performance**: Within the tested domains (toy problems, UCI regression datasets, and small-scale image classification), the method shows competitive performance.

### **Weaknesses**

**Theoretical Foundation**: The core mathematical construction in equation (4) is presented as general but only holds under specific loss function assumptions (strong/strict convexity). For cross-entropy loss used in classification experiments, equation (4) may not even represent valid solutions to optimization problem (3). This makes the theoretical analysis somewhat irrelevant to the larger-scale experimental validation.

**Theory-Practice Disconnect**: The meaningful deep learning experiments (classification on MNIST/CIFAR) operate entirely outside the theoretical framework. The remaining empirical evaluation consists only of nine UCI regression tasks and toy experiments, severely limiting the practical relevance of the work.

**Evaluation Methodology**: The paper introduces VMSP (Variance of Maximum Softmax Probability) as a UQ metric without explicit definition or proper theoretical justification (even taking into account appendix C).

**Limited Theoretical Novelty**: The core ideas of network linearization, NTK connections, and sample-then-optimize frameworks have been extensively explored. Particularly, Immer et al. (2021) already studied linearization, refinement through additional training, and NTK-GP connections. The contribution appears incremental rather than fundamental.

**Inadequate Experimental Scope**:

* Experiments limited to MNIST/CIFAR-scale datasets and outdated architectures rather than modern benchmarks and model architectures
* Missing evaluation on uncertainty-specific benchmarks (e.g., diabetic retinopathy detection tasks by Band (2022) or others)
* No comparison with important recent UQ methods, e.g. Immer et al. (2021).
* Related work discussion relegated entirely to appendix

### **Technical Concerns**

**Minor Missing Details**:

* Pseudo-inverse specification (left vs. right) not clarified
* Insufficient explanation of why objective perturbation (standard in sample-then-optimize?) is unnecessary
* Hyperparameter γ tuning may lead to validation overfitting without principled prior specification

**Methodological Issues**: The approach forgoes having a principled prior and predictive variance formulation, potentially making it more prone to overfitting on validation data compared to proper Bayesian methods.

**Scalability Limitations**: While computationally faster than deep ensembles, scalability to larger & more modern architectures remains undemonstrated, and the linearization approximation quality at scale is unclear.

---

Immer, Alexander, Maciej Korzepa, and Matthias Bauer. "Improving predictions of Bayesian neural nets via local linearization." International conference on artificial intelligence and statistics. PMLR, 2021\.

Band, Neil, et al. "Benchmarking bayesian deep learning on diabetic retinopathy detection tasks." arXiv preprint arXiv:2211.12717 (2022).

---

> ### Author Rebuttal · Authors · 2025-07-31
>
> We thank the reviewer for their feedback and constructive criticism. We hope that the following aids in alleviating your concerns.
>
> > The core mathematical construction in equation (4) is presented as general but only holds under specific loss function assumptions (strong/strict convexity)...The meaningful deep learning experiments (classification on MNIST/CIFAR) operate entirely outside the theoretical framework. The remaining empirical evaluation consists only of nine UCI regression tasks and toy experiments, severely limiting the practical relevance of the work...
>
> While our theory does not currently extend to cross-entropy loss, as is common in NTK-related work and many aspects of theory in machine learning more broadly, we would like to highlight the following two points. Firstly, it is practical extensions of the theory to settings where the assumptions fail that are often the most valuable to empirically test. As is seen in our results, we find that our framework also correctly quantifies the uncertainty in a range of image classification tasks. Secondly, we note that our theory does hold for classification using the Brier loss, and models trained using this loss have been shown to perform similarly to cross-entropy based models [1]. Testing NUQLS with the Brier loss is therefore expected to yield comparable performance and presents an interesting direction for future work. We have clarified these points in the updated version of our paper.
>
> > The authors...could better address the theory-practice disconnect and provide more analysis of when the linearization approximation might fail... How does the method perform when the linearization approximation quality degrades?
>
> We kindly refer the review to the _confidence intervals_ experimental results we have provided in our response to Reviewer Z6xt. We observe from the first given table that, in line with its theoretical construction, our method does not perform very well for poorly trained networks. This is because far away from a local minima, the linearization approximation may be poor. Further, we rely on the 'data-dependent' NTK that is learned during training [2]. While this is a space where PNC becomes a good method to choose, we do not view it as a limitation of our work. This is because (useful) networks in practice are well-trained; we observe in the second table of results that the performance of NUQLS is excellent once the network has been trained.
>
> > The paper introduces VMSP (Variance of Maximum Softmax Probability) as a UQ metric without explicit definition or proper theoretical justification (even taking into account appendix C).
>
> We are happy to provide a more explicit definition of VMSP: for a Bayesian method, we generally have a mean predictor $\mu : \mathbb{R}^d \to \mathbb{R}^c$ and a covariance function $\Sigma : \mathbb{R}^d \to \mathbb{R}^{c \times c}$, for example the mean and covariance of the linearized ensemble in the case of NUQLS, that output in the probit space. To compute VMSP for a given test point $\mathbf{x}^\star$, we first find $\hat{c} = \text{argmax}\_k ~\mu(\mathbf{x}^{*})\_{k}$, where $\mu(\mathbf{x}^\star)\_k$ denotes the $k$-th output of $\mu(\mathbf{x}^\star)$. That is, we find the class that the Bayesian method predicts, given $\mathbf{x}^\star$. We then define $\text{VMSP} := \Sigma(\mathbf{x}^\star)_{\hat{c},\hat{c}} = \sigma^2(\mathbf{x}^\star)\_{\hat{c}}$, that is, the variance of this prediction. We will include this description in the paper, to aid comprehension of the metric.
>
> As for theoretical justification, we can also be more explicit: we have detailed in Appendix C how the uncertainty of the predictions is captured by their variance. For a given dataset, we want a UQ method to provide low uncertainty for correctly predicted test points, and high uncertainty for incorrectly predicted or OOD test points. We then compare these distributions pictorially using a violin plot, and quantitatively using the median and skew values for the respective distributions. With the addition of a poorly-performing baseline model, we are able to easily compare the ability of UQ models to quantify uncertainty.
>
> > The core ideas of network linearization, NTK connections, and sample-then-optimize frameworks have been extensively explored. Particularly, Immer et al. (2021) already studied linearization, refinement through additional training, and NTK-GP connections. The contribution appears incremental rather than fundamental...No comparison with important recent UQ methods, e.g. Immer et al. (2021).
>
> To be clear, we have in fact compared to the work of Immer et al. (2021) (i.e. LLA) in most of the experiments, and kindly refer the reviewer to Appendix B.1, where we describe in detail the connection to, and differences with, the LLA framework methods. To summarize, the work of Immer et. al. showed how forming a Laplace Approximation over the parameter posterior of a linearized model resulted in a noisy NTK-GP predictive posterior. This approach incurs several challenges: forming the posterior requires explicitly forming and inverting the NTK gram matrix, independent GP's are required for each class in classification, and approximations such as last-layer and KFAC are required for scaling. In comparison, our method skips the parameter posterior, and uses the randomness in the null-space of the Jacobian to form a noise-free GP. Due to our derivation, we are able to easily sample from this GP posterior using linearized networks, without requiring covariance structure approximations. This enables us to out-perform LLA in larger image classification tasks (see Figure 3 in main text).
>
> > Experiments limited to MNIST/CIFAR-scale datasets and outdated architectures rather than modern benchmarks and model architectures...While computationally faster than deep ensembles... the linearization approximation quality at scale is unclear...Missing evaluation on uncertainty-specific benchmarks (e.g., diabetic retinopathy detection tasks by Band (2022) or others)...Can you demonstrate scalability to larger, more modern architectures? And additional datasets with additional epistemic uncertainty metrics?
>
> We thank the reviewer for raising this point, as the evaluation of our method on a larger problem will add to the value of our paper. We have now trained and evaluated NUQLS on ResNet50 trained on **ImageNet**, and using the ImageNet-o dataset for out-of-distribution testing [3]. We employed the pre-trained weights found on _torch.hub_ for ResNet50 / ImageNet. While we cannot provide an image of the violin plot of the VMSP, we can still provide the median values (where (>) denotes values which we want to be large, and (<) those we want to be small):
>
> |         | NUQLS   | BASE    |
> |---------|---------|---------|
> | IDC (<)      | $0.00$ | $0.0206$ |
> | IDIC (>)    | $0.232$ | $0.0197$ |
> | OOD (>)     | $0.100$ | $0.0197$ |
>
> and the skew values:
>
> |         | NUQLS   | BASE    |
> |---------|---------|---------|
> | IDC (>)    | $1.52$ | $1.21$ |
> | IDIC (<)   | $-0.88$ | $1.10$ |
> | OOD (<)    | $0.179$ | $0.985$ |
>
> For the in-distribution test set, we see excellent separation between the correct and incorrect predictions. As was stated in [4], approximately $21$\% of the images in ImageNet-o are in-distribution. Hence, we would expect that the OOD VMSP values to skew slightly more towards zero than the in-distribution incorrect predictions, as we observe. We note that ImageNet/ResNet50 is generally considered a large-scale setting for Bayesian methods.
>
> > Related work discussion relegated entirely to appendix
>
> We have relegated this discussion due to the large amount of related works, and the detail the discussion required. However, we can include a condensed version in the main body, given the additional space allowed in the camera-ready version.
>
> > Pseudo-inverse specification (left vs. right) not clarified
>
> As we assume $J_X$ to be full-row rank, the pseudo-inverse is computed as the right inverse.
>
> > Insufficient explanation of why objective perturbation (standard in sample-then-optimize?) is unnecessary
>
> We appreciate the opportunity to clarify this point, as this is a contribution of the paper. Standard sample-then-optimize works rely on randomness in the loss function. Instead, NUQLS relies on the fact that we are in an underdetermined linear system, and hence we can form an ensemble of solutions simply through the variability of random projections onto the null-space of the Jacobian. Thus, we do not require any variability in the loss function to sample from the posterior, in contrast to other sample-then-optimize methods.
>
> > The approach forgoes having a principled prior and predictive variance formulation, potentially making it more prone to overfitting on validation data compared to proper Bayesian methods... Hyperparameter $\gamma$ tuning may lead to validation overfitting without principled prior specification
>
> We respectfully disagree with the reviewer that we do not have a principled predictive variance formulation, as the predictive distribution follows a Gaussian Process. In regards the prior, we do not find evidence of overfitting on validation data, and instead show very competitive results. Further, we find that it is common for Bayesian methods to choose a Guassian isotropic prior.
>
> Please let us know if there are any other points that we can clarify, as we are eager to engage further to improve your assessment of our work.
>
> ---
> References:
> [1] Evaluation of Neural Architectures Trained with Square Loss vs Cross-Entropy in Classification Tasks, Hui et. al. 2020
>
> [2] Deep learning versus kernel learning: an empirical study of loss landscape geometry and the time evolution of the Neural Tangent Kernel, Fort et. al. 2020
>
> [3] Natural Adversarial Examples, Hendrycks et. al. 2017
>
> [4] In or Out? Fixing ImageNet Out-of-Distribution Detection Evaluation, Bitterwolf et. al. 2023

---

> ### Comment · Reviewer_j2t5 · 2025-08-04
>
> Thank you for the response. I'm increasing my score accordingly.
>
> PS: Have the authors seen Miani, Marco, Hrittik Roy, and Søren Hauberg. "Bayes without Underfitting: Fully Correlated Deep Learning Posteriors via Alternating Projections." arXiv preprint arXiv:2410.16901 (2024)? It also projects onto the kernel of the GGN to avoid underfitting.

---

> > ### Author Response · Authors · 2025-08-06
> >
> > We would like to thank the reviewer for increasing their score, and for providing the given reference. While we were familiar with the SLU method of [1] by some of the same team, we were not aware of the "Bayes without Underfitting" paper, and will take a closer look.
> >
> > ---
> > References:
> >
> > [1] Sketched Lanczos uncertainty score: a low-memory summary of the Fisher information,  Miani et. al. 2024

---

### Author Response · Authors · 2025-08-06
**Additional Results**

We will use this comment to present extra comparisons of our work to competing methods, as we discussed in our rebuttal.

Firstly, we provide a comparison of NUQLS vs BatchEnsembles (BE). We used the BE implementation from the code repository of the following paper [1]. As BE requires changes to the structure of the neural network (it is not post-hoc), we relied on the WideResNet (WRN) implementation from the paper. We trained a WRN-34-1 (approx. 1/2 million parameters) on FashionMNIST and used MNIST as the out-of-distribution dataset. For both BE and NUQLS, $10$ ensemble members were used, and both BE and the original network were trained with the same parameters. We find that while BE does well, NUQLS still outperforms it, whilst requiring less computational time (51mins 10secs for BE vs. 21mins for NUQLS + original network training on an A100 GPU). The median results are:

|               | NUQLS                  | BE                   | BASE    |
|---------------|------------------------|-----------------------|---------|
| WRN-34-1 FM: IDC (<)      | $0.00$            | $8.31 \times 10^{-5}$ | $0.0201$ |
| WRN-34-1 FM: IDIC (>)    | $0.178$            | $0.045$              | $0.020$ |
| WRN-34-1 FM: OOD (>)     | $0.173$            | $0.107$              | $0.020$ |

The skew values are:

|               | NUQLS                  | BE                  | BASE    |
|---------------|------------------------|-----------------------|---------|
| WRN-34-1 FM: IDC (>)      | $3.1$            | $2.8$                 | $1.03$ |
| WRN-34-1 FM: IDIC (<)    | $-0.342$           | $0.944$              | $1.13$ |
| WRN-34-1 FM: OOD (<)     | $-0.0875$           | $-0.272$            | $1.11$ |

We also compare NUQLS to Bayesian Deep Ensembles (BDE) on ResNet9 on FMNIST. This experiment required implementing BDE from the original paper [2], for which we closely followed the given implementation details as best as we could. We trained BDE using the same hyper-parameters as DE and used $10$ ensemble members. We see that BDE does not perform as well as NUQLS or DE in this case. The median results are:


|                | NUQLS                  | DE                    | BDE                    | SWAG                  | MC                    | LLA                    | BASE   |
|----------------|------------------------|-----------------------|------------------------|-----------------------|-----------------------|------------------------|--------|
| R9FM: IDC (<)  | $4.76 \times 10^{-15}$ | $1.31 \times 10^{-5}$ | $3.35 \times 10^{-20}$ | $4.64 \times 10^{-7}$ | $5.05 \times 10^{-7}$ | $9.02 \times 10^{-10}$ | $0.0206$ |
| R9FM: IDIC (>) | $0.182$                  | $0.0397$                | $0.00915$                | $0.0478 $               | $0.00413$               | $0.00372$                | $0.0199$ |
| R9FM: OOD (>)  | $0.217$                  | $0.109$                 | $0.0504$                 | $0.0748 $               | $0.00447$               | $0.0086$                 | $0.0199$ |

The skew values are:

|                | NUQLS | DE     | BDE     | SWAG   | MC      | LLA     | BASE   |
|----------------|-------|--------|---------|--------|---------|---------|--------|
| R9FM: IDC (>)  | $2.4$ | $3.51$  | $6.46$   | $3.72$  | $4.92$   | $6.46$   | $0.93$ |
| R9FM: IDIC (<) | $-0.615$ | $0.928$ | $1.43$ | $0.378$ | $1.6$ | $1.18$ | $1.15$ |
| R9FM: OOD (<)  | $-1.7$ | $0.321$  | $0.515$  | $0.0756$ | $1.69$ | $0.926$  | $1.05$ |

---
References:

[1] Encoding the latent posterior of Bayesian Neural Networks for uncertainty quantification, Franchi et. al. 2020

---

### Note · Authors · 2025-08-14

We are grateful for the sustained dialogue with the reviewers during the rebuttal period. We are confident we have addressed all concerns, and have made resulting modifications to the paper that will result in a more thoughtful, clear, and high-quality work.

The initial reviews had several suggestions to strengthen the quality of the paper: (1) including additional experimental evaluation to display the scalability of the method, (2) expanding baseline comparisons between NUQLS and other related methods, such as PNC, BE, SNGP, BDE, SGLD etc., (3) providing an explicit definition of the VMSP metric, and (4) clarifying when the linear approximation may fail, i.e. when the method would be expected to perform poorly. Additionally, several theoretical clarifications were requested, such as whether mode-collapse is a possibility, and the applicability of NUQLS to classification settings.

In response, we conducted a range of experimental evaluations. We tested NUQLS on the ImageNet dataset, using the ResNet50 model, a setting that is considered large-scale in the Bayesian UQ community. NUQLS was also evaluated on the *confidence intervals* experiment from the PNC paper. This experiment demonstrated that NUQLS outperforms PNC on trained, finite-width networks, while performance degrades on ultra-wide, untrained networks. This provided evidence of when NUQLS will fail; when the network is untrained, the linearization approximation is no longer appropriate. Comparison of NUQLS with BE, SNGP, BDE and SGLD on the VMSP metric was also provided; across the board, NUQLS delivered superior performance. Further, a more concrete and detailed explanation of VMSP was provided. Finally, we directly addressed all theoretical queries. For example, we provided justification for the use of NUQLS in classification settings, and a mathematical explanation for why mode-collapse cannot occur.

Overall, the feedback during the rebuttal phase was positive, and we found that reviewers were satisfied with the new experimental evidence, theoretical clarifications, and amendments to the manuscript. We believe that these changes have not only resolved the initial concerns but have also resulted in a more robust and impactful paper.

---

### Decision · Program_Chairs · 2025-09-17

**Decision:**

Accept (poster)

**Comment:**

This paper proposes a post-hoc, sampling-based uncertainty quantification (UQ) method for overparameterized networks at the end of training. The approach constructs efficient and meaningful deep ensembles by employing a (stochastic) gradient-descent sampling process on appropriately linearized networks.  Through a series of numerical experiments, the authors show that their method not only outperforms competing approaches in computational efficiency, but also maintains state-of-the-art performance across a variety of UQ metrics for both regression and classification tasks. The reviewers have stated that the method demonstrates significant computational speedups over deep ensembles. Moreover, it is simple and easy to implement, and under the tested domains is very competitive. They also say that the paper is clearly written and well-structured. Overall, this is a very interesting paper, and uncertainty quantification for neural networks is a key research topic. Therefore, I believe the field would benefit from this paper.